# Analysing Multi-Task Regression via Random Matrix Theory with Application to Time Series Forecasting

**Romain Ilbert**[*1, 2]    **Malik Tiomoko**[1]    **Cosme Louart**[3]    **Ambroise Odonnat**[1, 4]
**Vasilii Feofanov**[1]    **Themis Palpanas**[2]    **Ievgen Redko**[1]

[1]Huawei Noah's Ark Lab, Paris, France    [2]LIPADE, Paris Descartes University, Paris, France
[3] School of Data Science, The Chinese University of Hong Kong, Shenzhen, China
[4] Inria, Univ. Rennes 2, CNRS, IRISA

## Abstract

In this paper, we introduce a novel theoretical framework for multi-task regression, applying random matrix theory to provide precise performance estimations, under high-dimensional, non-Gaussian data distributions. We formulate a multi-task optimization problem as a regularization technique to enable single-task models to leverage multi-task learning information. We derive a closed-form solution for multi-task optimization in the context of linear models. Our analysis provides valuable insights by linking the multi-task learning performance to various model statistics such as raw data covariances, signal-generating hyperplanes, noise levels, as well as the size and number of datasets. We finally propose a consistent estimation of training and testing errors, thereby offering a robust foundation for hyperparameter optimization in multi-task regression scenarios. Experimental validations on both synthetic and real-world datasets in regression and multivariate time series forecasting demonstrate improvements on univariate models, incorporating our method into the training loss and thus leveraging multivariate information.

## 1   Introduction

The principle of multi-task learning, which encompasses concepts such as "learning to learn" and "knowledge transfer," offers an efficient paradigm that mirrors human intelligence. Unlike traditional machine learning, which tackles problems on a task-specific basis with tailored algorithms, multi-task learning leverages shared information across various tasks to enhance overall performance. This approach not only improves accuracy but also addresses challenges related to insufficient data and data cleaning. By analyzing diverse and multimodal data and structuring representations multi-task learning can significantly boost general intelligence, similar to how humans integrate skills. This concept is well-established [33] and has been successfully applied to a wide range of domains, such as computer vision [41], natural language processing [35, 37] and biology [17, 28, 42].

**Our Method.**    We consider the multi-task regression framework [5] with $T$ tasks with the input space $\mathcal{X}^{(t)} \subset \mathbb{R}^d$ and the output space $\mathcal{Y}^{(t)} \subset \mathbb{R}^q$ for $t \in \{1, \ldots, T\}$. For each task $t$, we assume that we are given $n_t$ training examples organized into the feature matrix $\mathbf{X}^{(t)} = [\mathbf{x}_1^{(t)}, \ldots, \mathbf{x}_{n_t}^{(t)}] \in \mathbb{R}^{d \times n_t}$ and the corresponding response matrix $\mathbf{Y}^{(t)} = [\mathbf{y}_1^{(t)}, \ldots, \mathbf{y}_{n_t}^{(t)}] \in \mathbb{R}^{q \times n_t}$, where $\mathbf{x}_i^{(t)} \in \mathcal{X}^{(t)}$ represents the $i$-th feature vector of the $t$-th task and $\mathbf{y}_i^{(t)} \in \mathcal{Y}^{(t)}$ is the associated response. In order to have a more tractable and insightful setup, we follow Romera-Paredes et al. [36] and consider the linear multi-task regression model. In particular, we study a straightforward linear signal-plus-noise model

---

*Correspondence to: romain.ilbert@hotmail.fr.

38th Conference on Neural Information Processing Systems (NeurIPS 2024).

that evaluates the response $\mathbf{y}_i^{(t)}$ for the $i$-th sample of the $t$-th task as follows:

$$\mathbf{Y}^{(t)} = \frac{\mathbf{X}^{(t)\top}\mathbf{W}_t}{\sqrt{Td}} + \boldsymbol{\varepsilon}^{(t)}, \quad \forall t \in \{1, \ldots, T\}, \tag{1}$$

where $\boldsymbol{\varepsilon}^{(t)} \in \mathbb{R}^{n_t \times q}$ is a matrix of noise vectors with each $\boldsymbol{\varepsilon}_i^{(t)} \sim \mathcal{N}(0, \boldsymbol{\Sigma}_N)$, $\boldsymbol{\Sigma}_N \in \mathbb{R}^{q \times q}$ denoting the covariance matrix. The matrix $\mathbf{W}_t \in \mathbb{R}^{d \times q}$ denotes the signal-generating hyperplane for task $t$. We denote the concatenation of all task-specific hyperplanes by $\mathbf{W} = [\mathbf{W}_1^\top, \ldots, \mathbf{W}_T^\top]^\top \in \mathbb{R}^{Td \times q}$. We assume that $\mathbf{W}_t$ can be decomposed as a sum of a common matrix $\mathbf{W}_0 \in \mathbb{R}^{d \times q}$, which captures the shared information across all the tasks, and a task-specific matrix $\mathbf{V}_t \in \mathbb{R}^{d \times q}$, which captures deviations specific to the task $t$:

$$\mathbf{W}_t = \mathbf{W}_0 + \mathbf{V}_t. \tag{2}$$

Given the multitask regression framework and the linear signal-plus-noise model, we now want to retrieve the common and specific hyperplanes, $\mathbf{W}_0$ and $\mathbf{V}_t$, respectively. To achieve this, we study the following minimization problem governed by a parameter $\lambda$ that controls the balance between the common and specific components of $\mathbf{W}$:

$$\mathbf{W}_0^*, \{\mathbf{V}_t^*\}_{t=1}^T, \lambda^* = \arg\min \quad \frac{1}{2\lambda}\|\mathbf{W}_0\|_F^2 + \frac{1}{2}\sum_{t=1}^T \frac{\|\mathbf{V}_t\|_F^2}{\gamma_t} + \frac{1}{2}\sum_{t=1}^T \left\| \mathbf{Y}^{(t)} - \frac{\mathbf{X}^{(t)\top}\mathbf{W}_t}{\sqrt{Td}} \right\|_F^2. \tag{3}$$

where $\gamma = [\gamma_1, \ldots, \gamma_T]$ is a vector of task-specific regularization hyperparameters. Each $\gamma_t$ controls how much the model overfits (small $\gamma_t$) or underfits (large $\gamma_t$) on task $t$.

**Contributions and Main Results.** We seek to provide a rigorous theoretical study of (3) in high dimension, providing practical insights that make the theoretical application implementable in practice. Our contributions can be summarized in four major points:

1. We provide exact train and test risks for the solutions of (3) using random matrix theory. We then decompose the test risk into a signal and noise term responsible for the effectiveness of multi-task learning and the negative transfer, respectively.

2. We show how the signal and noise terms compete with each other depending on $\lambda$ for which we obtain an optimal value optimizing the test risk.

3. In a particular case, we derive a closed-form data-dependent solution for the optimal $\lambda^\star$, involving raw data covariances, signal-generating hyperplane, noise levels, and the size and number of data sets.

4. We offer empirical estimates of these model variables to develop a ready-to-use method for hyperparameter optimization in multitask regression problems within our framework.

**Applications.** As an application, we view multivariate time series forecasting (MTSF) as a multi-task problem and show how the proposed regularization method can be used to efficiently employ univariate models in the multivariate case. Firstly, we demonstrate that our method improves `PatchTST` [32] and `DLinear` [53] compared to the case when these approaches are applied independently to each variable. Secondly, we show that the univariate models enhanced by the proposed regularization achieve performance similar to the current state-of-the-art multivariate forecasting models such as `SAMformer` [19] and `iTransformer` [24].

## 2 Related Work

**Usual Bounds in Multi-Task Learning.** Since its introduction [1, 5, 26], multi-task learning (MTL) has demonstrated that transfer learning concepts can effectively leverage shared information to solve related tasks simultaneously. Recent research has shifted focus from specific algorithms or minimization problems to deriving the lowest achievable risk given an MTL data model. The benefits of transfer learning are highlighted through risk bounds, using concepts such as contrasts [23], which study high-dimensional linear regression with auxiliary samples characterized by the sparsity of their contrast vector or transfer distances [29], which provide lower bounds for target generalization

error based on the number of labeled source and target data and their similarities. Additionally, task-correlation matrices [30] have been used to capture the relationships inherent in the data model.

However, these studies remain largely theoretical, lacking practical algorithms to achieve the proposed bounds. They provide broad estimates and orders of magnitude that, while useful for gauging the feasibility of certain objectives, do not offer precise performance evaluations. Moreover, these estimates do not facilitate the optimal tuning of hyperparameters within the MTL framework, a critical aspect of our study. We demonstrate the importance of precise performance evaluations and effective hyperparameter tuning to enhance the practical application of MTL.

**Random Matrix Theory in Multi-Task Learning.** In the high-dimensional regime, Random Matrix Theory allows obtaining precise theoretical estimates on functionals (e.g., train or test risk) of data matrices with the number of samples comparable to data dimension [2, 13, 31, 46]. In the multi-task classification case, large-dimensional analysis has been performed for Least Squares SVM [48] and supervised PCA [47] showing counterproductivity of tackling tasks independently. We inspire our optimization strategy from [48] and conduct theoretical analysis for the multi-task regression, which introduces unique theoretical challenges and with a different intuition since we need to consider another data generation model and a regression learning objective. Our data modeling shares similarity with [52] which theoretically studied conditions of a negative transfer under covariate and model shifts. Our work focuses on the selection of hyperparameters within a general optimization framework, considering both a hyperparameter $\lambda$ to relate all tasks and specific parameters $\gamma_t$ to balance overfitting and underfitting within each task, accommodating non-Gaussian distributions and employing recent developments such as deterministic equivalents and concentration inequalities.

**High dimensional analysis for regression.** Regression is an important problem that has been extensively explored in the context of single-task learning for high-dimensional data, employing tools such as Random Matrix Theory [12], physical statistics [9, 15], and the Convex Gaussian MinMax Theorem (CGMT) [43], among others. Typically, authors employ a linear signal plus noise model to calculate the asymptotic test risk as a function of the signal-generating parameter and the covariance of the noise. Our research builds upon these studies, extending them to a multi-task learning framework. In doing so, we derive several insights unique to multi-task learning. However, none of these previous studies offer a practical method for estimating the asymptotic test risk, which is dependent on the ground truth signal-generating parameter. Therefore, our work not only extends previous studies into the multi-task case within a generic data distribution (under the assumption of a concentrated random vector), but also provides a practical method for estimating these quantities.

## 3 Framework

**Notation.** Throughout this study, matrices are represented by bold uppercase letters (e.g., matrix $\mathbf{A}$), vectors by bold lowercase letters (e.g., vector $\mathbf{v}$), and scalars by regular, non-bold typeface (e.g., scalar $a$). The notation $\mathbf{A} \otimes \mathbf{B}$ for matrices or vectors $\mathbf{A}, \mathbf{B}$ is the Kronecker product. $\mathcal{D}_{\mathbf{x}}$ or $\mathrm{Diag}(\mathbf{x})$ stands for a diagonal matrix containing on its diagonal the elements of the vector $\mathbf{x}$. The superscripts $t$ and $i$ are used to denote the task and the sample number, respectively, e.g., $\mathbf{x}_i^{(t)}$ writes the $i$-th sample of the $t$-th task. The canonical vector of $\mathbb{R}^T$ is denoted by $\mathbf{e}_t^{[T]}$ with $[e_t^{[T]}]_i = \delta_{ti}$. Given a matrix $M \in \mathbb{R}^{p \times n}$, the Frobenius norm of $\mathbf{M}$ is denoted $\|\mathbf{M}\|_F \equiv \sqrt{\mathrm{tr}(\mathbf{M}^\top \mathbf{M})}$. For our theoretical analysis, we introduce the following notation of training data:

$$\mathbf{Y} = [\mathbf{Y}^{(1)}, \dots, \mathbf{Y}^{(T)}] \in \mathbb{R}^{q \times n}, \qquad \mathbf{Z} = \sum_{t=1}^{T} \left( \mathbf{e}_t^{[T]} \mathbf{e}_t^{[T]\top} \right) \otimes \mathbf{X}^{(t)} \in \mathbb{R}^{Td \times n}$$

where $n = \sum_{t=1}^{T} n_t$ is the total number of samples in all the tasks.

### 3.1 Regression Model

We define $\mathbf{V} = [\mathbf{V}_1^\top, \dots, \mathbf{V}_T^\top]^\top \in \mathbb{R}^{dT \times q}$ and propose to solve the linear multi-task problem by finding $\hat{\mathbf{W}} = [\hat{\mathbf{W}}_1^\top, \dots, \hat{\mathbf{W}}_T^\top]^\top \in \mathbb{R}^{dT \times q}$ which solves the following optimization problem under the assumption of relatedness between the tasks, i.e., $\mathbf{W}_t = \mathbf{W}_0 + \mathbf{V}_t$ for all tasks $t$:

$$\min_{(\mathbf{W}_0, \mathbf{V}) \in \mathbb{R}^{d \times q} \times \mathbb{R}^{dT \times q}} \mathcal{J}(\mathbf{W}_0, \mathbf{V}) \tag{4}$$

where

$$\mathcal{J}(\mathbf{W}_0, \mathbf{V}) \equiv \frac{1}{2\lambda}\|\mathbf{W}_0\|_F^2 + \frac{1}{2}\sum_{t=1}^{T}\frac{\|\mathbf{V}_t\|_F^2}{\gamma_t} + \frac{1}{2}\sum_{t=1}^{T}\left\|\mathbf{Y}^{(t)} - \frac{\mathbf{X}^{(t)\top}\mathbf{W}_t}{\sqrt{Td}}\right\|_F^2.$$

The objective function consists of three components: a regularization term for $\mathbf{W}_0$ to mitigate overfitting, a task-specific regularization term that controls deviations $\mathbf{V}_t$ from the shared weight matrix $\mathbf{W}_0$, and a loss term quantifying the error between the predicted outputs and the actual responses for each task.

The optimization problem is convex for any positive values of $\lambda, \gamma_1, \ldots, \gamma_T$, and it possesses a unique solution. The details of the calculus are given in Appendix B. The formula for $\hat{\mathbf{W}}_t$ is given as follows:

$$\hat{\mathbf{W}}_t = \left(\mathbf{e}_t^{[T]\top} \otimes \mathbf{I}_d\right)\frac{\mathbf{AZQY}}{\sqrt{Td}}, \qquad \hat{\mathbf{W}}_0 = \left(\mathbb{1}_T^\top \otimes \lambda\mathbf{I}_d\right)\frac{\mathbf{ZQY}}{\sqrt{Td}},$$

where with $\mathbf{Q} = \left(\frac{\mathbf{Z}^\top\mathbf{AZ}}{Td} + \mathbf{I}_n\right)^{-1} \in \mathbb{R}^{n\times n}$, and $\mathbf{A} = \left(\mathcal{D}_{\boldsymbol{\gamma}} + \lambda\mathbb{1}_T\mathbb{1}_T^\top\right) \otimes \mathbf{I}_d \in \mathbb{R}^{Td\times Td}$.

### 3.2 Assumptions

In order to use Random Matrix Theory (RMT) tools, we make two assumptions on the data distribution and the asymptotic regime. Following [50], we adopt a concentration hypothesis on the feature vectors $\mathbf{x}_i^{(t)}$, which was shown to be highly effective for analyzing machine learning problems [11, 14]. The justification and implications of these assumptions can be found in appendix A.2 and A.3.

**Assumption 1** (Concentrated Random Vector). *We assume that there exists two constants $C, c > 0$ (independent of dimension $d$) such that, for any task $t$, for any 1-Lipschitz function $f : \mathbb{R}^d \to \mathbb{R}$, any feature vector $\mathbf{x}^{(t)} \in \mathcal{X}^{(t)}$ verifies:*

$$\forall t > 0 : \mathbb{P}(|f(\mathbf{x}^{(t)}) - \mathbb{E}[f(\mathbf{x}^{(t)})]| \geq t) \leq Ce^{-(t/c)^2},$$

$$\mathbb{E}[\mathbf{x}^{(t)}] = 0 \quad \text{and} \quad \text{Cov}[\mathbf{x}^{(t)}] = \mathbf{\Sigma}^{(t)}.$$

In particular, we distinguish the following scenarios: $\mathbf{x}_i^{(t)} \in \mathbb{R}^d$ are concentrated when they are (i) independent Gaussian random vectors with covariance of bounded norm, (ii) independent random vectors uniformly distributed on the $\mathbb{R}^d$ sphere of radius $\sqrt{d}$, and most importantly (iii) any Lipschitz transformation $\phi(\mathbf{x}_i^{(t)})$ of the above two cases, with bounded Lipschitz norm. Scenario (iii) is especially pertinent for modeling data in realistic settings. Recent research [39] has demonstrated that images produced by generative adversarial networks (GANs) are inherently qualified as concentrated random vectors.

Next, we present a classical RMT assumption that establishes a commensurable relationship between the number of samples and dimension.

**Assumption 2** (High-dimensional asymptotics). *As $d \to \infty$, $n_t = \mathcal{O}(d)$ and $T = \mathcal{O}(1)$. More specifically, we assume that $n/d \xrightarrow{a.s.} c_0 < \infty$ with $n = \sum_{t=1}^{T} n_t$.*

Although different from classical asymptotic where the number of samples is implicitly assumed to be exponentially larger than the dimension, the high-dimensional asymptotic finds many applications including telecommunications [10], finance [34] and machine learning [11, 14, 48].

## 4 Main Theoretical Results

### 4.1 Estimation of the Performances

Given the training dataset $\mathbf{X} \in \mathbb{R}^{n\times d}$ with the corresponding response variable $\mathbf{Y} \in \mathbb{R}^{n\times q}$ and for any test dataset $\mathbf{x}^{(t)} \in \mathbb{R}^d$ and the corresponding response variable $\mathbf{y}^{(t)} \in \mathbb{R}^q$, we aim to derive the theoretical training and test risks defined as follows:

$$\mathcal{R}_{train}^\infty = \frac{1}{Tn}\mathbb{E}\left[\|\mathbf{Y} - g(\mathbf{X})\|_2^2\right], \quad \mathcal{R}_{test}^\infty = \frac{1}{T}\sum_{t=1}^{T}\mathbb{E}[\|\mathbf{y}^{(t)} - g(\mathbf{x}^{(t)})\|_2^2]$$

The output score $g(\mathbf{x}^{(t)}) \in \mathbb{R}^q$ for data $\mathbf{x}^{(t)} \in \mathbb{R}^d$ of task $t$ is given by:

$$g(\mathbf{x}^{(t)}) = \frac{1}{\sqrt{Td}} \left( \mathbf{e}_t^{[T]} \otimes \mathbf{x}^{(t)} \right)^\top \hat{\mathbf{W}}_t = \frac{1}{Td} \left( \mathbf{e}_t^{[T]} \otimes \mathbf{x}^{(t)} \right)^\top \mathbf{AZQY} \tag{5}$$

In particular, the output for the training score is given by:

$$g(\mathbf{X}) = \frac{1}{Td} \mathbf{Z}^\top \mathbf{AZQY}, \qquad \mathbf{Q} = \left( \frac{\mathbf{Z}^\top \mathbf{AZ}}{Td} + \mathbf{I}_{Td} \right)^{-1} \tag{6}$$

To understand the statistical properties of $\mathcal{R}_{train}^\infty$ and $\mathcal{R}_{test}^\infty$ for the linear generative model, we study the statistical behavior of the resolvent matrix $\mathbf{Q}$ defined in (6). The notion of a deterministic equivalent from random matrix theory [16] is particularly useful here as it allows us to design a deterministic matrix equivalent to $\mathbf{Q}$ in the sense of linear forms. Specifically, a deterministic equivalent $\bar{\mathbf{M}}$ of a random matrix $\mathbf{M}$ is a deterministic matrix that satisfies $u(\mathbf{M} - \bar{\mathbf{M}}) \to 0$ almost surely for any bounded linear form $u : \mathbb{R}^{d \times d} \to \mathbb{R}$. We denote this as $\mathbf{M} \leftrightarrow \bar{\mathbf{M}}$. This concept is crucial to our analysis because we need to estimate quantities such as $\frac{1}{d} \mathrm{tr}\,(\mathbf{AQ})$ or $\mathbf{a}^\top \mathbf{Qb}$, which are bounded linear forms of $\mathbf{Q}$ with $\mathbf{A}, \mathbf{a}$ and $\mathbf{b}$ all having bounded norms. For convenience, we further express some contributions of the performances with the so-called "coresolvent" defined as $\tilde{\mathbf{Q}} \equiv (\frac{\mathbf{A}^{\frac{1}{2}} \mathbf{ZZ}^T \mathbf{A}^{\frac{1}{2}}}{Td} + \mathbf{I}_{Td})^{-1}$.

Using Lemma 1 provided in the Appendix C.1, whose proofs are included in Appendices C.2, C.3 and C.4, we establish the deterministic equivalents that allow us to introduce our Theorems 5 and 1, characterizing the asymptotic behavior of both training and testing risks.

**Theorem 1** (Asymptotic train and test risk). *Assuming that the training data vectors $\mathbf{x}_i^{(t)}$ and the test data vectors $\mathbf{x}^{(t)}$ are concentrated random vectors, and given the growth rate assumption (Assumption 2), it follows that:*

$$\mathcal{R}_{test}^\infty = \underbrace{\frac{\mathrm{tr}\left( \mathbf{W}^\top \mathbf{A}^{-\frac{1}{2}} \bar{\tilde{\mathbf{Q}}}_2(\mathbf{A}) \mathbf{A}^{-\frac{1}{2}} \mathbf{W} \right)}{Td}}_{\text{signal term}} + \underbrace{\frac{\mathrm{tr}(\boldsymbol{\Sigma}_n \bar{\mathbf{Q}}_2)}{Td} + \mathrm{tr}\,(\boldsymbol{\Sigma}_n)}_{\text{noise terms}}. \tag{ATR}$$

*In addition, the asymptotic risk on the training data is given by*

$$\mathcal{R}_{train}^\infty \leftrightarrow \frac{\mathrm{tr}\left( \mathbf{W}^\top \mathbf{A}^{-1/2} \bar{\tilde{\mathbf{Q}}} \mathbf{A}^{-1/2} \mathbf{W} \right)}{Tn} - \frac{\mathrm{tr}\left( \mathbf{W}^\top \mathbf{A}^{-1/2} \bar{\tilde{\mathbf{Q}}}_2(\mathbf{I}_{Td}) \mathbf{A}^{-1/2} \mathbf{W} \right)}{Tn} + \frac{\mathrm{tr}\left( \boldsymbol{\Sigma}_n \bar{\mathbf{Q}}_2 \right)}{Tn},$$

*where $\bar{\tilde{\mathbf{Q}}}$, $\bar{\tilde{\mathbf{Q}}}_2(\mathbf{I}_{Td})$ and $\bar{\mathbf{Q}}_2$ are respectively deterministic equivalents for $\tilde{\mathbf{Q}}$, $\tilde{\mathbf{Q}}^2$ and $\mathbf{Q}^2$.*

We defer the full proof of this theorem to Appendix D and provide an outline of the test risk proof below. Note that derivations of the asymptotic risk for the training data are more complex and involve computing deterministic equivalents (Lemma 1 of Appendix C.1), which is a powerful tool commonly used in Random Matrix Theory. In the derived theorem, we achieve a stronger convergence result compared to the almost surely convergence. Specifically, we prove a concentration result for the training and test risk with an exponential convergence rate for sufficiently large values of $n$ and $d$.

$$
\begin{aligned}
\mathcal{R}_{test}^{\infty} &= \frac{1}{T}\mathbb{E}[\|\mathbf{y} - g(\mathbf{x})\|_2^2] \\
&= \frac{1}{T}\mathbb{E}\left[\|\frac{\mathbf{x}^\top \mathbf{W}}{\sqrt{Td}} + \varepsilon - \frac{\mathbf{x}^\top \mathbf{AZQZ}^\top \mathbf{W}}{Td\sqrt{Td}} - \frac{\mathbf{x}^\top \mathbf{AZQ}\varepsilon}{Td}\|_2^2\right] \\
&= \frac{1}{T}\mathbb{E}\left[\frac{\operatorname{tr}\left(\mathbf{W}^\top \boldsymbol{\Sigma}\mathbf{W}\right)}{Td} - \frac{2\operatorname{tr}\left(\mathbf{W}^\top \boldsymbol{\Sigma}\mathbf{AZQZ}^\top\mathbf{W}\right)}{(Td)^2} + \operatorname{tr}\left(\varepsilon^\top\varepsilon\right) + \frac{\operatorname{tr}\left(\mathbf{W}^\top \mathbf{ZQZ}^\top \mathbf{A}\boldsymbol{\Sigma}\mathbf{AZQZ}^\top\mathbf{W}\right)}{(Td)^3} + \right. \\
&\qquad \left. \frac{\operatorname{tr}\left(\varepsilon^\top \mathbf{QZ}^\top \mathbf{A}\boldsymbol{\Sigma}\mathbf{AZQ}\varepsilon\right)}{(Td)^2}\right] \\
&= \frac{\operatorname{tr}\left(\mathbf{W}^\top \boldsymbol{\Sigma}\mathbf{W}\right)}{Td} - 2\frac{\operatorname{tr}\left(\mathbf{W}^\top \boldsymbol{\Sigma}\mathbf{A}^{\frac{1}{2}}(\mathbf{I}_{Td} - \tilde{\mathbf{Q}})\mathbf{A}^{-\frac{1}{2}}\mathbf{W}\right)}{Td} + \operatorname{tr}\left(\varepsilon^\top\varepsilon\right) + \frac{\operatorname{tr}\left(\mathbf{W}^\top \boldsymbol{\Sigma}\mathbf{W}\right)}{Td} \\
&\qquad - 2\frac{\operatorname{tr}\left(\mathbf{W}^\top \boldsymbol{\Sigma}A^{\frac{1}{2}}\bar{\tilde{\mathbf{Q}}}A^{-\frac{1}{2}}\mathbf{W}\right)}{Td} + \frac{\mathbf{W}^\top \mathbf{A}^{-\frac{1}{2}}\bar{\tilde{\mathbf{Q}}}_2(\mathbf{A})\mathbf{A}^{-\frac{1}{2}}\mathbf{W}}{Td} + \frac{1}{Td}\operatorname{tr}(\boldsymbol{\Sigma}_N\bar{\mathbf{Q}}_2) + O\left(\frac{1}{\sqrt{d}}\right)
\end{aligned}
$$

## 4.2 Error Contribution Analysis

To gain theoretical insights, we analyze (ATR) consisting of the signal and the noise components.

**Signal Term.** The signal term can be further approximated, up to some constants as $\operatorname{tr}(\mathbf{W}^\top(\mathbf{A}\boldsymbol{\Sigma} + \mathbf{I})^{-2}\mathbf{W})$ with $\boldsymbol{\Sigma} = \sum_{t=1}^{T}\frac{n_t}{d}\boldsymbol{\Sigma}^{(t)}$. The matrix $(\mathbf{A}\boldsymbol{\Sigma} + \mathbf{I})^{-2}$ plays a crucial role in amplifying the signal term $\operatorname{tr}(\mathbf{W}^\top\mathbf{W})$, which in turn allows the test risk to decrease. The off-diagonal elements of $\mathbf{A}\boldsymbol{\Sigma} + \mathbf{I})^{-2}$ amplify the cross terms ($\operatorname{tr}(\mathbf{W}_v^\top\mathbf{W}_t)$ for $t \neq v$), enhancing the multi-task aspect, while the diagonal elements amplify the independent terms ($\|\mathbf{W}_t\|_2^2$). This structure is significant in determining the effectiveness of multi-task learning.

Furthermore, both terms decrease with an increasing number of samples $n_t$, smaller values of $\gamma_t$, and a larger value of $\lambda$. The cross term, which is crucial for multi-task learning, depends on the matrix $\boldsymbol{\Sigma}_t^{-1}\boldsymbol{\Sigma}_v$. This matrix represents the shift in covariates between tasks. When the features are aligned (i.e., $\boldsymbol{\Sigma}_t^{-1}\boldsymbol{\Sigma}_v = \mathbf{I}_d$), the cross term is maximized, enhancing multi-task learning. However, a larger Fisher distance between the covariances of the tasks results in less favorable correlations for multi-task learning.

**Noise term.** Similar to the signal term, the noise term can be approximated, up to some constants, as $\operatorname{tr}(\boldsymbol{\Sigma}_N(\mathbf{A}^{-1} + \boldsymbol{\Sigma})^{-1})$. However, there is a major difference between the way both terms are expressed in the test risk. The noise term does not include any cross terms because the noise across different tasks is independent. In this context, only the diagonal elements of the matrix are significant. This diagonal term increases with the sample size and the value of $\lambda$. It is responsible for what is known as negative transfer. As the diagonal term increases, it negatively affects the transfer of learning from one task to another. This is a critical aspect to consider in multi-task learning scenarios.

## 4.3 Simplified Model for Clear Insights

In this section, we specialize the theoretical analysis to the simple case of two tasks ($T = 2$). We assume that the tasks share the same identity covariance and that $\gamma_1 = \gamma_2 \equiv \gamma$. Under these conditions, the test risk can be approximated, up to some constants, as

$$
\mathcal{R}_{test}^{\infty} = \mathbf{D}_{IL}\left(\|\mathbf{W}_1\|_2^2 + \|\mathbf{W}_2\|_2^2\right) + \mathbf{C}_{MTL}\mathbf{W}_1^\top\mathbf{W}_2 + \mathbf{N}_{NT}\operatorname{tr}\boldsymbol{\Sigma}_n
$$

where the diagonal term (independent learning) $\mathbf{D}_{IL}$, the cross term (multi-task learning) $\mathbf{C}_{MTL}$, and the noise term (negative transfer) $\mathbf{N}_{NT}$ have closed-form expressions depending on $\gamma$ and $\lambda$:

$$
\mathbf{D}_{IL} = \frac{(c_0(\lambda + \gamma) + 1)^2 + c_0^2\lambda^2}{(c_0(\lambda + \gamma) + 1)^2 - c_0^2\lambda^2}, \quad \mathbf{C}_{MTL} = \frac{-2c_0\lambda(c_0(\lambda + \gamma) + 1)}{(c_0(\lambda + \gamma) + 1)^2 - c_0^2\lambda^2}
$$

$$
\mathbf{N}_{NT} = \frac{(c_0(\lambda + \gamma)^2 + (\lambda + \gamma) - c_0\lambda^2)^2 + \lambda^2}{\left((c_0(\lambda + \gamma) + 1)^2 - c_0^2\lambda^2\right)^2}
$$

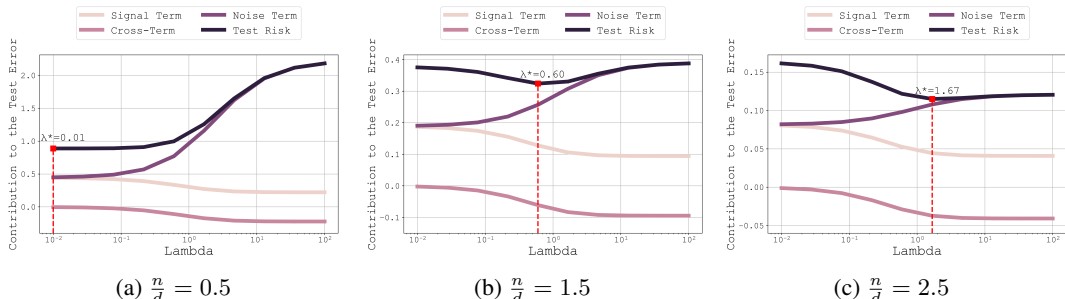

Figure 1: Test loss contributions $\mathbf{D}_{IL}$, $\mathbf{C}_{MTL}$, $\mathbf{N}_{NT}$ across three sample size regimes. Test risk exhibits decreasing, increasing, or convex shapes based on the regime. $\lambda^*$ from theory are marked.

We recall that $c_0$ has been defined in the Assumption 2. As previously mentioned, the test risk is primarily composed of two terms: the signal term and the noise term, which are in competition with each other. The more similar the tasks are, the stronger the signal term becomes. In the following plot, we illustrate how this competition can influence the risk. Depending on the value of the parameter $\lambda$ and the sample sizes, the risk can either decrease monotonically, increase monotonically, or exhibit a convex behavior. This competition can lead to an optimal value for $\lambda$, which interestingly has a simple closed-form expression that can be obtained by deriving the $\mathcal{R}_{test}^{\infty}$ w.r.t. $\lambda$ as follows (see details in Appendix E.4):

$$\lambda^\star = \frac{n}{d}SNR - \frac{\gamma}{2}, \text{ with } SNR = \frac{\|\mathbf{W}_1\|_2^2 + \|\mathbf{W}_2\|_2^2}{\text{tr}\mathbf{\Sigma}_N} + \frac{\mathbf{W}_1^\top \mathbf{W}_2}{\text{tr}\mathbf{\Sigma}_N}.$$

### 4.4 Comparison between Empirical and Theoretical Predictions

In this section, we compare the theoretical predictions with the empirical results. Our experiment is based on a two-task setting ($T = 2$) defined as $\mathbf{W}_1 \sim \mathcal{N}(0, I_p)$ with $\mathbf{W}_2 = \alpha\mathbf{W}_1 + \sqrt{1 - \alpha^2}\mathbf{W}_1^\perp$. $\mathbf{W}_1^\perp$ represents any vector orthogonal to $\mathbf{W}_1$ and $\alpha \in [0, 1]$. This setting allows us to adjust the similarity between tasks through $\alpha$.

Figure 2 shows a comparison of the theoretical and empirical classification errors for different values of $\lambda$, highlighting the error-minimizing value of $\lambda$. Despite the relatively small values of $n$ and $p$, there is a very precise match between the asymptotic theory and the practical experiment. This is particularly evident in the accurate estimation of the optimal value for $\lambda$.

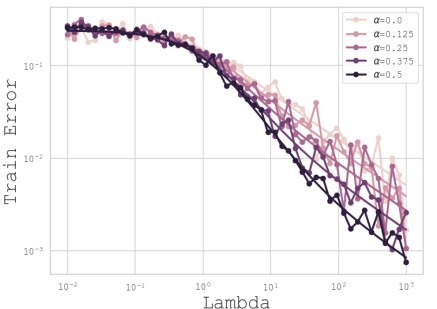 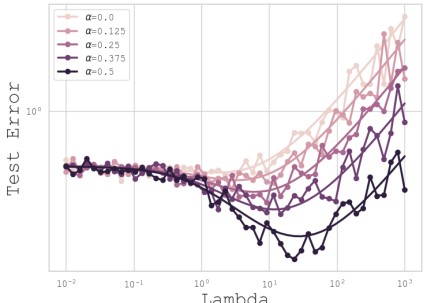

Figure 2: Empirical and theoretical train and test MSE as functions of the parameter $\lambda$ for different values of $\alpha$. The smooth curves represent the theoretical predictions, while the corresponding curves with the same color show the empirical results, highlighting that the empirical observations indeed match the theoretical predictions.

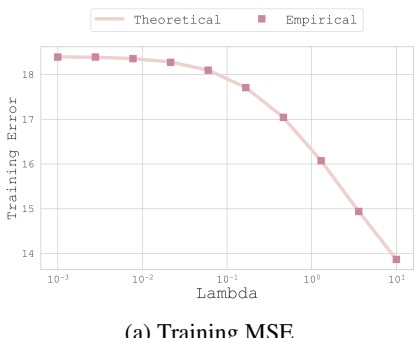

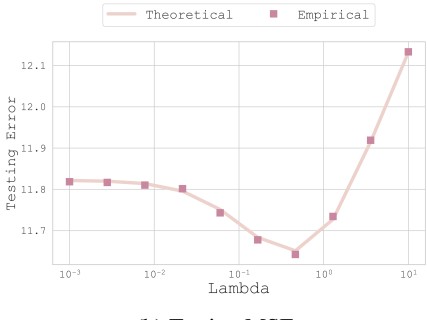

|(a) Training MSE|(b) Testing MSE|

Figure 3: Theoretical vs Empirical MSE as function of regularization parameter $\lambda$. Close fit between the theoretical and the empirical predictions which underscores the robustness of the theory in light of varying assumptions as well as the accuracy of the suggested estimates. We consider the first two channels as the the two tasks and $d = 144$. 95 samples are used for the training and 42 samples are used for the test.

## 5 Hyperparameter Optimization in Practice

### 5.1 Empirical Estimation of Task-wise Signal, Cross Signal, and Noise

All the quantities defined in Theorem 1 are known except for the bilinear expressions $\frac{1}{Td}\mathrm{tr}(\mathbf{W}^\top \mathbf{MW})$ and $\frac{1}{Td}\mathrm{tr}\left(\mathbf{\Sigma}_N \mathbf{M}\right)$. These quantities can be consistently estimated under Assumptions 2 as follows (see details in Section F of the Appendix) :

$$\frac{1}{Td}\mathrm{tr}\left(\mathbf{W}^\top \mathbf{MW}\right) - \zeta(\mathbf{M}) \xrightarrow{\text{a.s.}} 0, \qquad \frac{1}{Td}\mathrm{tr}\left(\mathbf{\Sigma}_N \mathbf{M}\right) - \hat{\sigma}\mathrm{tr}\mathbf{M} \xrightarrow{\text{a.s.}} 0$$

where $\zeta(\mathbf{M}) = \frac{1}{Td}\mathrm{tr}(\hat{\mathbf{W}}^\top \kappa^{-1}(\mathbf{M})\mathbf{M}\hat{\mathbf{W}}) - \frac{\hat{\sigma}}{Td}\mathrm{tr}\bar{\bar{\mathbf{Q}}}_2(\mathbf{A}^{\frac{1}{2}}\kappa^{-1}(\mathbf{M})\mathbf{A}^{\frac{1}{2}})$.

We define the estimate of the noise as $\hat{\sigma} = \lim\limits_{\substack{\lambda \to 0 \\ \gamma_t \to \infty}} \mathcal{R}_{train}^\infty$ and the function $\kappa^{-1}$ is the functional inverse of the mapping $\kappa : \mathbb{R}^{Td \times Td} \to \mathbb{R}^{q \times q}$ defined as follows

$$\kappa(\mathbf{M}) = \mathbf{M} - 2\mathbf{A}^{-\frac{1}{2}}\bar{\bar{\mathbf{Q}}}\mathbf{A}^{\frac{1}{2}}\mathbf{M} + \mathbf{A}^{-\frac{1}{2}}\bar{\bar{\mathbf{Q}}}_2(\mathbf{A}^{\frac{1}{2}}\mathbf{MA}^{\frac{1}{2}})\mathbf{A}^{-\frac{1}{2}} - \frac{n}{(Td)^2}\mathbf{A}^{-\frac{1}{2}}\mathbf{\Sigma}\mathbf{A}^{-\frac{1}{2}}\bar{\bar{\mathbf{Q}}}_2(\mathbf{A}^{\frac{1}{2}}\mathbf{MA}^{\frac{1}{2}}).$$

### 5.2 Application to Multi-task Regression

In our study, we apply the theoretical framework presented in our paper to a real-world regression problem, specifically, the *Appliance Energy dataset* which aims to predict the total usage of a house. This dataset is a multivariate regression dataset containing 138 time series each of dimension 24. We select two of these features as 2 tasks to cast the problem as a multi-task learning regression problem.

Figure 3 presents a comparison between the theoretical predictions and empirical outcomes of our proposed linear framework. Despite the assumptions made in the main body of the paper, the theoretical predictions closely align with the experimental results. This demonstrates that our estimates are effective in practice and provide a reliable approximation of the optimal regularization.

In essence, our theoretical framework, when applied to real-world multi-task learning regression problems, yields practical and accurate estimates, thereby validating its effectiveness and applicability.

### 5.3 Relevance of the theoretical insights beyond the case of linear models

While non-linear models are widely used, establishing their theoretical foundations is challenging. Therefore, we focused on linear models, which, despite their simplicity, provide valuable insights into more complex models.

Our results show that test risk curves for non-linear models follow patterns predicted by our theory. This is expected because non-linear models in time series forecasting typically use a linear output layer for prediction. Thus, we can apply our theory to the inputs of this final linear layer. This approach is valid due to data concentration and the Lipschitz nature of neural networks, ensuring outputs of the non-linear part don't deviate significantly from the inputs.

Moreover, multivariate time series models often treat channels separately using univariate methods, missing cross-channel information. Our results in Section 5.4 show that our method surpasses univariate baselines by optimally regularizing with $\lambda$ and $\gamma$, supporting our theory's applicability to non-linear models as the final linear layer effectively leverages concentrated inputs.

Finally, our regularization approach differs from traditional cross-task regularizations that use one task per dataset. We consider each prediction as a task and introduce $\gamma_t$ parameters alongside $\lambda$. These parameters enforce multivariate regularization and control underfitting or overfitting per task. This method is tractable since it's applied at the model's final layer.

The similarity between curves for non-linear and linear models indicates our findings are robust; non-linear models also exhibit optimal regularization parameters, enhancing performance in multivariate forecasting.

## 5.4 Application to Multivariate Time Series Forecasting

Our theoretical framework is applied in the context of Multivariate Time Series Forecasting, with related work detailed in Appendix G.1. We previously applied this framework in a linear setting, and now aim to evaluate its empirical validity in the non-linear setting of neural networks. The results presented in this section represent the best test MSE, assuming the ability to find the optimal lambda value, which can be considered as an oracle scenario. A study of these limitations can be found in Appendix H.

**Motivation.** Our approach is applied to the MTSF setting for several reasons. Firstly, many models currently used are essentially univariate, where predictions for individual series are simply concatenated without exploiting the multivariate information inherent in traditional benchmarks. Given that these benchmarks are designed for multivariate forecasting, leveraging multivariate information should yield better results. Secondly, our theoretical framework can benefit this domain, as most predictive models use a linear layer on top of the model to project historical data of length $d$ for predicting the output of length $q$. This characteristic aligns well with our method, making it a promising fit for enhancing forecasting accuracy.

**Our approach.** We propose a novel method for MTSF by modifying the loss function to incorporate both individual feature transformations $f_t$ and a shared transformation $f_0$. Each univariate-specific transformation $f_t$ is designed to capture the unique dynamics of its respective feature, while $f_0$ serves as a common transformation applied across all features to capture underlying patterns shared among them. We consider a neural network $f$ with inputs $\mathbf{X} = [\mathbf{X}^{(1)}, \mathbf{X}^{(2)}, \ldots, \mathbf{X}^{(T)}]$, where $T$ is the number of channels and $\mathbf{X}^{(t)} \in \mathbb{R}^{n \times d}$. For a univariate model without MTL regularization, we predict $\mathbf{Y} = [\mathbf{Y}^{(1)}, \mathbf{Y}^{(2)}, \ldots, \mathbf{Y}^{(T)}] = [f_1(\mathbf{X}^{(1)}), \ldots, f_T(\mathbf{X}^{(T)})]$ and $\mathbf{Y}^{(t)} \in \mathbb{R}^{n \times q}$. We compare these models with their corresponding versions that include MTL regularization, formulated as: $f_t^{MTL}(\mathbf{X}^{(t)}) = f_t(\mathbf{X}^{(t)}) + f_0(\mathbf{Y})$ with $f_t : \mathbb{R}^{n \times d} \to \mathbb{R}^{n \times q}$ and $f_0 : \mathbb{R}^{n \times qT} \to \mathbb{R}^{n \times q}$. We define our regularized loss as follows:

$$\mathcal{L}(\mathbf{X}, \mathbf{Y}) = \sum_{t=1}^{T} \|\mathbf{Y}^{(t)} - f_t^{MTL}(\mathbf{X}^{(t)})\|_F^2 + \lambda \|f_0(\mathbf{X})\|_F^2 + \sum_{t=1}^{T} \gamma_t \|f_t(\mathbf{X}^{(t)}\|_F^2, \quad \forall t \in \{1, \ldots, T\}.$$

where $\mathbf{Y}^{(t)}$ are the true predictions, $f_t$ represents the univariate model for each channel $t$, and $\lambda$ is our regularization parameter, for which we have established a closed form in the case of linear $f_t$. $f_0$ serves a role equivalent to $W_0$, which was defined in our theoretical study and allows for the regularization of the common part. This component can be added at the top of a univariate model. The parameters $\gamma_t$ enable the regularization of the specialized parts $f_t$.

In our setup, $f_t$ is computed in a similar way as in the model without regularization and $f_0$ is computed by first flattening the concatenation of the predictions of $\mathbf{X}^{(t)}$, then applying a linear projection leveraging common multivariate information before reshaping. The loss function is specifically designed to balance fitting the multivariate series using $f_0$ and the specific channels using $f_t$. This

Table 1: MTL regularization results. Algorithms marked with $^\dagger$ are state-of-the-art multivariate models and serve as baseline comparisons. All others are univariate. We compared the models with MTL regularization to their corresponding versions without regularization. Each MSE value is derived from 3 different random seeds. MSE values marked with * indicate that the model with MTL regularization performed significantly better than its version without regularization, according to a Student's t-test with a p-value of 0.05. MSE values are in **bold** when they are the best in their row, indicating the top-performing models.

| Dataset | $H$ | with MTL regularization | | | without MTL regularization | | | | | |
|---|---|---|---|---|---|---|---|---|---|---|
| | | PatchTST | DLinearU | Transformer | PatchTST | DLinearU | DLinearM | Transformer | SAMformer$^\dagger$ | iTransformer$^\dagger$ |
| ETTh1 | 96 | 0.385 | **0.367**$^*$ | 0.368 | 0.387 | 0.397 | 0.386 | 0.370 | 0.381 | 0.386 |
| | 192 | 0.422 | **0.405**$^*$ | 0.407$^*$ | 0.424 | 0.422 | 0.437 | 0.411 | 0.409 | 0.441 |
| | 336 | 0.433$^*$ | 0.431 | 0.433 | 0.442 | 0.431 | 0.481 | 0.437 | **0.423** | 0.487 |
| | 720 | 0.430$^*$ | 0.454 | 0.455$^*$ | 0.451 | 0.428 | 0.519 | 0.470 | **0.427** | 0.503 |
| ETTh2 | 96 | 0.291 | **0.267**$^*$ | 0.270 | 0.295 | 0.294 | 0.333 | 0.273 | 0.295 | 0.297 |
| | 192 | 0.346$^*$ | **0.331**$^*$ | 0.337 | 0.351 | 0.361 | 0.477 | 0.339 | 0.340 | 0.380 |
| | 336 | **0.332**$^*$ | 0.367 | 0.366$^*$ | 0.342 | 0.361 | 0.594 | 0.369 | 0.350 | 0.428 |
| | 720 | **0.384**$^*$ | 0.412 | 0.405$^*$ | 0.393 | 0.395 | 0.831 | 0.428 | 0.391 | 0.427 |
| Weather | 96 | **0.148** | 0.149$^*$ | 0.154$^*$ | 0.149 | 0.196 | 0.196 | 0.170 | 0.197 | 0.174 |
| | 192 | **0.190** | 0.206$^*$ | 0.198$^*$ | 0.193 | 0.243 | 0.237 | 0.214 | 0.235 | 0.221 |
| | 336 | **0.242**$^*$ | 0.249$^*$ | 0.258 | 0.246 | 0.283 | 0.283 | 0.260 | 0.276 | 0.278 |
| | 720 | **0.316**$^*$ | 0.326$^*$ | 0.331 | 0.322 | 0.339 | 0.345 | 0.326 | 0.334 | 0.358 |

approach enhances the model's generalization across various forecasting horizons and datasets. More details on our regularized loss function can be found in Appendix G.2

**Results.** We present experimental results on different forecasting horizons, using common benchmark MTSF datasets, the characteristics of which are outlined in the Appendix G.3. Our models include PatchTST [32], known to be on par with state-of-the-art in MTSF while being a univariate model, a univariate DLinear version called DLinearU compared to its multivariate counterpart DLinearM [53], and a univariate Transformer [19] with temporal-wise attention compared to the multivariate state-of-the-art models SAMformer [19] and iTransformer [24] . Table 1 provides a detailed comparison of the test mean squared errors (MSE) for different MTSF models, emphasizing the impact of MTL regularization. Models with MTL regularization are compared to their versions without regularization, as well as SAMformer and iTransformer. All the experiments can be found in Appendix G.4.

Adding MTL regularization improves the performance of PatchTST, DLinearU, and Transformer in most cases. When compared to state-of-the-art multivariate models, the MTL-regularized models are often competitive. SAMformer is outperformed by at least one MTL-regularized method per horizon and dataset, except for ETTh1 with horizons of 336 and 720. iTransformer is consistently outperformed by at least one MTL-regularized methods regardless of the dataset and horizon.

The best performing methods are PatchTST and DLinearU with MTL regularization. These models not only outperform their non-regularized counterparts, often significantly as shown by Student's t-tests with a p-value of 0.05, but also surpass state-of-the-art multivariate models like SAMformer and Transformer. This superior performance is indicated by the bold values in the table.

Finally, MTL regularization enhances the performance of univariate models, making them often competitive with state-of-the-art multivariate methods like SAMformer and iTransformer. This approach seems to better captures shared dynamics among tasks, leading to more accurate forecasts.

## 6 Conclusions and Future Works

In this article, we have explored linear multi-task learning by deriving a closed-form solution for an optimization problem that capitalizes on the wealth of information available across multiple tasks. Leveraging Random Matrix Theory, we have been able to obtain the asymptotic training and testing risks, and have proposed several insights into high-dimensional multi-task learning regression. Our theoretical analysis, though based on a simplified model, has been effectively applied to multi-task regression and multivariate forecasting, using both synthetic and real-world datasets. We believe that our work lays a solid foundation for future research, paving the way for using random matrix theory with more complex models, such as deep neural networks, within the multi-task learning framework.

## Acknowledgements

Work partially funded by EU project AI4Europe (101070000).

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

# Appendix

**Roadmap.** This appendix provides the technical details omitted in the main paper. It starts with an overview of the setup considered in the paper in **Section A**. **Section B** offers a detail computation for $\hat{\mathbf{W}}_t$ and $\hat{\mathbf{W}}_0$. **Section C** contains a proof of Lemma 1. **Section D** explains the theoretical steps for deriving the training and test risks, as well as the deterministic equivalents. **Section E** discusses the technical tools used to derive the main intuitions presented by the theory. **Section F** focuses on the derivation of the estimations of the main quantities involved in the training and test risks. **Section G** complements the experimental study by presenting additional experiments. Finally, **Section H** deals with the limitations of our approach in a non-linear setting.

## Table of Contents

# A   Multi-task Learning setup

## A.1   On the zero-mean assumption

We would like to note that we are assuming that both the noise $\boldsymbol{\varepsilon}$ and the feature $\mathbf{x}_i^{(t)}$ have zero mean. This is a common assumption in many statistical models and it simplifies the analysis. However, this assumption is not restrictive. In practice, if the data or the response variable are not centered, we can always preprocess the data by subtracting the mean. This preprocessing step brings us back to the zero-mean setting that we consider in our theoretical analysis.

## A.2   On the Assumption 1

Data are concentrated random vectors, meaning high-dimensional data maintain stable under complex (Lipschitz) transformations. The strong performance of neural networks on tasks like image recognition and NLP suggests that these models produce stable predictions. As Lipschitz transformations, they maintain controlled distances between inputs, ensuring stability. Recent studies have demonstrated and experimentally confirmed that both real-world data and synthetically generated data using GANs exhibit concentration properties, supporting this assumption. This makes our assumption more realistic than traditional Gaussian assumptions, as it does not rely on specific hypotheses about the shape of the data distribution, but rather on the stability of statistical properties after transformation. Consequently, analyzing a framework of concentrated random vectors is more theoretically challenging than using Gaussian assumptions and represents a key novelty of our theory.

## A.3   On the Assumption 2

The dimension $d$ is of the same order of magnitude as the sample size $n$. This joint growth captures data complexity better than assuming a fixed feature size with increasing samples, which can oversimplify models. Our theory works for fixed $d$ and $n$, unbiased by specific parameter choices. The accuracy of empirical predictions depends on both $d$ and $n$, with variance scaling as $O\left(\frac{1}{\sqrt{dn}}\right)$. Larger $d$ and $n$ reduce variance, making empirical results more reliable and closer to theoretical values. Conversely, smaller $d$ and $n$ increase variance, affecting single predictions. However, this scaling is still better than $n$ growing indefinitely with fixed $d$, where variance scales as $O\left(\frac{1}{\sqrt{n}}\right)$, leading to increased bias. Implication of the assumptions. In Section 5.2, Figure 3, the Appliance Energy dataset illustrates our assumptions' realism. Despite the moderate sample size of 42 samples with 142 dimensions and non-synthetic data, the theoretical curve fits the empirical predictions well. Additionally, with synthetic data ($d = 100$, $n = 100$), the theory matches the empirical curve across various hyperparameters (Figure 2), showcasing the predictive power of Random Matrix Theory.

# B   Minimization Problem

## B.1   Computation of $\hat{\mathbf{W}}_t$ and $\hat{\mathbf{W}}_0$

The proposed multi task regression finds $\hat{\mathbf{W}} = [\hat{\mathbf{W}}_1^\top, \ldots, \hat{\mathbf{W}}_k^\top]^\top \in \mathbb{R}^{dT \times q}$ which solves the following optimization problem using the additional assumption of relatedness between the tasks ($\mathbf{W}_t = \mathbf{W}_0 + \mathbf{V}_t$ for all tasks $t$):

$$\min_{(\mathbf{W}_0, \mathbf{V}) \in \mathbb{R}^d \times \mathbb{R}^{d \times T} \times \mathbb{R}^T} \mathcal{J}(\mathbf{W}_0, \mathbf{V}) \tag{7}$$

where

$$\mathcal{J}(\mathbf{W}_0, \mathbf{V}) \equiv \frac{1}{2\lambda}\text{tr}\left(\mathbf{W}_0^\top \mathbf{W}_0\right) + \frac{1}{2}\sum_{t=1}^{T}\frac{\text{tr}\left(\mathbf{V}_t^\top \mathbf{V}_t\right)}{\gamma_t} + \frac{1}{2}\sum_{t=1}^{T}\text{tr}\left(\boldsymbol{\xi}_t^\top \boldsymbol{\xi}_t\right)$$

$$\boldsymbol{\xi}_t = \mathbf{Y}^{(t)} - \frac{\mathbf{X}^{(t)\top}\mathbf{W}_t}{\sqrt{Td}}, \quad \forall t \in \{1, \ldots, T\}.$$

The Lagrangian introducing the lagrangian parameters for each task $t$, $\boldsymbol{\alpha}_t \in \mathbb{R}^{n_t \times q}$ reads as

$$\mathcal{L}(\mathbf{W}_0, \mathbf{V}_t, \boldsymbol{\xi}_t, \boldsymbol{\alpha}_t) = \frac{1}{2\lambda}\text{tr}\left(\mathbf{W}_0^\top \mathbf{W}_0\right) + \frac{1}{2}\sum_{t=1}^{T}\frac{\text{tr}\left(\mathbf{V}_t^\top \mathbf{V}_t\right)}{\gamma_t} + \frac{1}{2}\sum_{t=1}^{T}\text{tr}\left(\boldsymbol{\xi}_t^\top \boldsymbol{\xi}_t\right)$$
$$+ \sum_{t=1}^{T}\text{tr}\left(\boldsymbol{\alpha}_t^\top\left(\mathbf{Y}^{(t)} - \frac{\mathbf{X}^{(t)\top}(\mathbf{W}_0 + \mathbf{V}_t)}{\sqrt{Td}} - \boldsymbol{\xi}_t\right)\right)$$

Differentiating with respect to the unknown variables $\hat{\mathbf{W}}_0$, $\hat{\mathbf{V}}_t$, $\boldsymbol{\xi}_t$, $\boldsymbol{\alpha}_t$ and $\mathbf{b}_t$, we get the following system of equation

$$\frac{1}{\lambda}\hat{\mathbf{W}}_0 - \sum_{t=1}^{T}\frac{\mathbf{X}^{(t)}\boldsymbol{\alpha}_t}{\sqrt{Td}} = 0$$

$$\frac{1}{\gamma_t}\hat{\mathbf{V}}_t - \frac{\mathbf{X}^{(t)}\boldsymbol{\alpha}_t}{\sqrt{Td}} = 0$$

$$\boldsymbol{\xi}_t - \boldsymbol{\alpha}_t = 0$$

$$\mathbf{Y}^{(t)} - \frac{\mathbf{X}^{(t)\top}\hat{\mathbf{W}}_0}{\sqrt{Td}} - \frac{\mathbf{X}^{(t)\top}\hat{\mathbf{V}}_t}{\sqrt{Td}} - \boldsymbol{\xi}_t = 0$$

Plugging the expression of $\hat{\mathbf{W}}_0$, $\hat{\mathbf{V}}_t$ and $\boldsymbol{\xi}_t$ into the expression of $\mathbf{Y}^{(t)}$ gives

$$\mathbf{Y}^{(t)} = \lambda\sum_{t=1}^{T}\frac{\mathbf{X}^{(t)\top}\mathbf{X}^{(t)}}{Td}\boldsymbol{\alpha}_t + \gamma_t\frac{\mathbf{X}^{(t)\top}\mathbf{X}^{(t)}}{Td}\boldsymbol{\alpha}_t + \boldsymbol{\alpha}_t$$

which can be rewritten as

$$\mathbf{Y}^{(t)} = (\lambda + \gamma_t)\frac{\mathbf{X}^{(t)\top}\mathbf{X}^{(t)}}{Td}\boldsymbol{\alpha}_t + \lambda\sum_{v\neq t}\frac{\mathbf{X}^{(t)\top}\mathbf{X}^{(v)}}{Td}\boldsymbol{\alpha}_v + \boldsymbol{\alpha}_t$$

With $\mathbf{Y} = [\mathbf{Y}^{(1)\top}, \ldots, \mathbf{Y}^{(T)\top}]^\top \in \mathbb{R}^{n \times q}$, $\boldsymbol{\alpha} = [\boldsymbol{\alpha}_1^\top, \ldots, \boldsymbol{\alpha}_k^\top]^\top \in \mathbb{R}^{n \times q}$, $\mathbf{Z} = \sum_{t=1}^{T}\mathbf{e}_t^{[T]}\mathbf{e}_t^{[T]\top} \otimes \mathbf{X}^{(t)} \in \mathbb{R}^{Td \times n}$, this system of equations can be written under the following compact matrix form:

$$\mathbf{Q}^{-1}\boldsymbol{\alpha} = \mathbf{Y}$$

with $\mathbf{Q} = \left(\frac{\mathbf{Z}^\top \mathbf{A}\mathbf{Z}}{Td} + \mathbf{I}_n\right)^{-1} \in \mathbb{R}^{n \times n}$, and $\mathbf{A} = \left(\mathcal{D}_{\boldsymbol{\gamma}} + \lambda\mathbb{1}_T\mathbb{1}_T^\top\right) \otimes \mathbf{I}_d \in \mathbb{R}^{Td \times Td}$.

Solving for $\boldsymbol{\alpha}$ then gives:

$$\boldsymbol{\alpha} = \mathbf{Q}\mathbf{Y}$$

Moreover, using $\hat{\mathbf{W}}_t = \hat{\mathbf{W}}_0 + \hat{\mathbf{V}}_t$, the expression of $\mathbf{W}_t$ becomes:

$$\hat{\mathbf{W}}_t = \left(\mathbf{e}_t^{[T]\top} \otimes \mathbf{I}_d\right)\frac{\mathbf{A}\mathbf{Z}\boldsymbol{\alpha}}{\sqrt{Td}},$$

$$\hat{\mathbf{W}}_0 = \left(\mathbb{1}_T^\top \otimes \lambda\mathbf{I}_d\right)\frac{\mathbf{Z}\boldsymbol{\alpha}}{\sqrt{Td}}.$$

## C Lemma 1 and proof with Random Matrix Theory

### C.1 Lemma 1

**Lemma 1** (Deterministic equivalents for $\tilde{\mathbf{Q}}$, $\tilde{\mathbf{Q}}\mathbf{M}\tilde{\mathbf{Q}}$ and $\mathbf{Q}^2$ for any $\mathbf{M} \in \mathbb{R}^{n\times n}$). *Under the concentrated random vector assumption for each feature vector $\mathbf{x}_i^{(t)}$ and under the growth rate assumption (Assumption 2), for any deterministic $\mathbf{M} \in \mathbb{R}^{n\times n}$, we have the following convergence:*

$$\tilde{\mathbf{Q}} \leftrightarrow \bar{\tilde{\mathbf{Q}}}, \qquad \tilde{\mathbf{Q}}\mathbf{M}\tilde{\mathbf{Q}} \leftrightarrow \bar{\tilde{\mathbf{Q}}}_2(\mathbf{M}), \qquad \mathbf{Q}^2 \leftrightarrow \bar{\mathbf{Q}}_2$$

*where $\bar{\tilde{\mathbf{Q}}}_2$, $\bar{\tilde{\mathbf{Q}}}$ and $\bar{\mathbf{Q}}_2$ are defined as follows*

$$\bar{\tilde{\mathbf{Q}}} = \left( \sum_{t=1}^T \frac{c_0 \mathbf{C}^{(t)}}{1 + \delta_t} + \mathbf{I}_{Td} \right)^{-1}, \quad \delta_t = \frac{1}{Td}\mathrm{tr}\left( \mathbf{\Sigma}^{(t)}\bar{\tilde{\mathbf{Q}}} \right), \quad \mathbf{C}^{(t)} = \mathbf{A}^{\frac{1}{2}}\left( \mathbf{e}_t^{[T]} \otimes \mathbf{\Sigma}^{(t)} \right)\mathbf{A}^{\frac{1}{2}}$$

$$\bar{\tilde{\mathbf{Q}}}_2(\mathbf{M}) = \bar{\tilde{\mathbf{Q}}}\mathbf{M}\bar{\tilde{\mathbf{Q}}} + \frac{1}{Td}\sum_{t=1}^T \frac{d_t}{1+\delta_t}\bar{\tilde{\mathbf{Q}}}\mathbf{C}^{(t)}\bar{\tilde{\mathbf{Q}}}, \qquad \mathbf{d} = \left( \mathbf{I}_T - \frac{1}{Td}\mathbf{\Psi} \right)^{-1}\mathbf{\Psi}(\mathbf{M}) \in \mathbb{R}^T$$

$$\bar{\mathbf{Q}}_2 = \mathbf{I}_n - Diag_{t\in[T]}(v_t \mathbf{I}_{n_t}), \quad v_t = \frac{1}{Td}\frac{\mathrm{tr}(\mathbf{C}^{(t)}\bar{\tilde{\mathbf{Q}}})}{(1+\delta_t)^2} + \frac{1}{Td}\frac{\mathrm{tr}\left( \mathbf{C}^{(t)}\bar{\tilde{\mathbf{Q}}}_2(\mathbf{I}_n) \right)}{(1+\delta_t)^2}$$

*where*

$$\Psi(M) = \left( \frac{n_t}{Td}\frac{\mathrm{tr}\left( \mathbf{C}^{(t)}\bar{\tilde{\mathbf{Q}}}\mathbf{M}\bar{\tilde{\mathbf{Q}}} \right)}{1+\delta_t} \right)_{t\in[T]} \in \mathbb{R}^T, \qquad \mathbf{\Psi} = \left( \frac{n_t}{Td}\frac{\mathrm{tr}\left( \mathbf{C}^{(t)}\bar{\tilde{\mathbf{Q}}}\mathbf{C}^{(t')}\bar{\tilde{\mathbf{Q}}} \right)}{(1+\delta_t)(1+\delta_{t'})} \right)_{t,t'\in[T]} \in \mathbb{R}^{T\times T},$$

### C.2 Deterministic equivalent of the resolvent $\tilde{\mathbf{Q}}$

The evaluation of the expectation of linear forms on $\tilde{\mathbf{Q}}$ and $\tilde{\mathbf{Q}}^2$ can be found in the literature. To find a result that meets exactly our setting, we will cite [25] that is a bit more general since it treats cases where $\mathbb{E}[x_i^{(t)}] \neq 0$ for $t \in [T]$ and $i \in [n_t]$. Unlike the main paper, and to be more general, the study presented below is "quasi asymptotic" meaning that the results are true for finite value of $d, n$. Let us first rewrite the general required hypotheses, adapting them to our setting. For that purpose, we consider in the rest of this paper a certain asymptotic $I \subset \{(d,n), d \in \mathbb{N}, n \in \mathbb{N}\} = \mathbb{N}^2$ satisfying:

$$\{d, \exists n \in \mathbb{N} : (d,n) \in I\} = \mathbb{N} \qquad \text{and} \qquad \{n, \exists d \in \mathbb{N} : (d,n) \in I\} = \mathbb{N}.$$

such that $n$ and $d$ can tend to $\infty$ but with some constraint that is given in the first item of Assumption 3 below. Given two sequences $(a_{d,n})_{d,n\in I}, (b_{d,n})_{d,n\in I} > 0$, the notation $a_{d,n} \leq O(b_{d,n})$ (or $a \leq O(b)$) means that there exists a constant $C > 0$ such that for all $(d,n) \in I$, $a_{d,n} \leq Cb_{d,n}$.

**Assumption 3.** *There exists some constants $C, c > 0$ independent such that:*

- $n \leq O(d)$

- $\mathbf{Z} = (\mathbf{z}_1, \ldots, \mathbf{z}_n) \in \mathbb{R}^{Td\times n}$ *has independent columns*

- *for any $(d,n) \in I$, and any $f : \mathbb{R}^{Td\times n} \to \mathbb{R}$ 1-Lipschitz for the euclidean norm:*

$$\mathbb{P}\left( |f(\mathbf{Z}) - \mathbb{E}[f(\mathbf{Z})]| \geq t \right) \leq Ce^{-ct^2}.$$

- $\forall i \in \{n, \exists d \in \mathbb{N}, (d,n) \in I\}$: $\|\mathbb{E}[\mathbf{z}_i]\| \leq O(1)$.

**Theorem 2** ([25], Theorem 0.9.). *Given $T \in \mathbb{N}$, $\mathbf{Z} \in \mathbb{R}^{Td\times n}$ and two deterministic $A \in \mathbb{R}^{Td\times Td}$, we note $\tilde{\mathbf{Q}} \equiv (\frac{1}{Td}\mathbf{A}^{\frac{1}{2}}\mathbf{Z}\mathbf{Z}^\top\mathbf{A}^{\frac{1}{2}} + I_{Td})^{-1}$. If $\mathbf{Z}$ satisfies Assumption 3 and $\mathbf{M} \in \mathbb{R}^{Td\times Td}$ is a deterministic matrix satisfying $\|\mathbf{M}\|_F \leq 1$, one has the concentration:*

$$\mathbb{P}\left( \left| \mathrm{tr}(M\tilde{\mathbf{Q}}) - \mathrm{tr}(M\bar{\tilde{\mathbf{Q}}}_{\delta(\mathbf{S})}(\mathbf{S})) \right| \geq t \right) \leq Ce^{-ct^2},$$

where $\mathbf{S} = (\mathbf{S}_1, \dots, \mathbf{S}_n) = (\mathbb{E}[\mathbf{z}_1 \mathbf{z}_1^\top], \dots, \mathbb{E}[\mathbf{z}_n \mathbf{z}_n^\top])$, for $\boldsymbol{\delta} \in \mathbb{R}^n$, $\bar{\bar{\mathbf{Q}}}_{\boldsymbol{\delta}}$ is defined as:

$$\bar{\bar{\mathbf{Q}}}_{\boldsymbol{\delta}}(\mathbf{S}) = \left( \frac{1}{Td} \sum_{i \in [n]} \frac{\mathbf{A}^{\frac{1}{2}} \mathbf{S}_i \mathbf{A}^{\frac{1}{2}}}{1 + \boldsymbol{\delta}_i} + I_{Td} \right)^{-1},$$

and $\boldsymbol{\delta}(\mathbf{S})$ is the unique solution to the system of equations:

$$\forall i \in [n]: \quad \boldsymbol{\delta}(\mathbf{S})_i = \frac{1}{n} \operatorname{tr} \left( \mathbf{A}^{\frac{1}{2}} \mathbf{S}_i \mathbf{A}^{\frac{1}{2}} \bar{\bar{\mathbf{Q}}}_{\boldsymbol{\delta}(\mathbf{S})} \right).$$

We end this subsection with some results that will be useful for next subsection on the estimation of bilinear forms on $\tilde{\mathbf{Q}}$.

**Lemma 2** ([25], Lemmas 4.2, 4.6). *Under the setting of Theorem 2, given a deterministic vector* $\mathbf{u} \in \mathbb{R}^{Td}$ *such that* $\|\mathbf{u}\| \le O(1)$ *and two deterministic matrices* $\mathbf{U}, \mathbf{V}$ *such that* $\|\mathbf{U}\|, \|\mathbf{V}\| \le O(1)$ *and a power* $r > 0$, $r \le O(1)$:

- $\mathbb{E}\left[ \left| \mathbf{u}^\top \mathbf{U} \tilde{\mathbf{Q}}_{-i} \mathbf{V} \mathbf{z}_i \right|^r \right] \le O(1)$

- $\mathbb{E}\left[ \left| \frac{1}{Td} \mathbf{z}_i^\top \mathbf{U} \tilde{\mathbf{Q}}_{-i} \mathbf{V} \mathbf{z}_i - \mathbb{E}\left[ \frac{1}{Td} \operatorname{tr}\left( \Sigma_i \mathbf{U} \bar{\bar{\mathbf{Q}}} \mathbf{B} \right) \right] \right|^r \right] \le O\left( \frac{1}{d^{\frac{r}{2}}} \right).$

### C.3 Deterministic equivalent of bilinear forms of the resolvent

To simplify the expression of the following theorem, we take $\mathbf{A} = \mathbf{I}_{Td}$. One can replace $\mathbf{Z}$ with $\mathbf{A}^{\frac{1}{2}} \mathbf{Z}$ to retrieve the result necessary for the main paper.

**Theorem 3.** *Under the setting of Theorem 2, with* $\mathbf{A} = \mathbf{I}_{Td}$, *one can estimate for any deterministic matrices* $\mathbf{U}, \mathbf{V} \in \mathbb{R}^{Td}$ *such that* $\|\mathbf{U}\|, \|\mathbf{V}\| \le O(1)$ *and any deterministic vector* $\mathbf{u}, \mathbf{v} \in \mathbb{R}^{Td}$ *such that* $\|\mathbf{u}\|, \|\mathbf{v}\| \le 1$, *if one notes* $\mathbf{B} = \frac{1}{Td} \mathbf{V}$ *or* $\mathbf{B} = \mathbf{u} \mathbf{v}^\top$, *one can estimate:*

$$\left| \mathbb{E}\left[ \operatorname{tr}(\mathbf{B} \tilde{\mathbf{Q}} \mathbf{U} \tilde{\mathbf{Q}}) \right] - \Psi(\mathbf{U}, \mathbf{B}) - \frac{1}{Td} \Psi(\mathbf{U})^\top \left( \mathbf{I}_n - \frac{1}{Td} \Psi \right)^{-1} \Psi(\mathbf{B}) \right| \le O\left( \frac{1}{\sqrt{d}} \right) \qquad (8)$$

*where we noted:*

- $\bar{\bar{\mathbf{Q}}} \equiv \bar{\bar{\mathbf{Q}}}_{\boldsymbol{\delta}}(\mathbf{S})$, $\boldsymbol{\delta} = \boldsymbol{\delta}(\mathbf{S})$,

- $\Psi \equiv \frac{1}{Td} \left( \frac{\operatorname{tr}\left( \mathbf{S}_i \bar{\bar{\mathbf{Q}}} \mathbf{S}_j \bar{\bar{\mathbf{Q}}} \right)}{(1+\delta_i)(1+\delta_j)} \right)_{i,j \in [n]} \in \mathbb{R}^{n,n}$

- $\forall \mathbf{U} \in \mathbb{R}^{n \times n} : \Psi(\mathbf{U}) \equiv \frac{1}{Td} \left( \frac{\operatorname{tr}\left( \mathbf{U} \bar{\bar{\mathbf{Q}}} \mathbf{s}_i \bar{\bar{\mathbf{Q}}} \right)}{1+\delta_i} \right)_{i \in [n]} \in \mathbb{R}^n$

- $\forall \mathbf{U}, \mathbf{V} \in \mathbb{R}^{n \times n} : \Psi(\mathbf{U}, \mathbf{V}) \equiv \frac{1}{Td} \operatorname{tr}\left( \mathbf{U} \bar{\bar{\mathbf{Q}}} \mathbf{V} \bar{\bar{\mathbf{Q}}} \right) \in \mathbb{R}$

If there exist $T < n$ dinstinct matrices $\mathbf{C}_1, \dots, \mathbf{C}_T$ such that:

$$\{ \mathbf{S}_1, \dots, \mathbf{S}_n \} = \{ \mathbf{C}_1, \dots, \mathbf{C}_T \},$$

and if we denote $\forall t \in [T]\ n_t = \#\{ i \in [n] \mid \mathbf{S}_i = \mathbf{C}_t \}$ and:

$$P \equiv \left( I_T - \left( \frac{n_t n_v}{(Td)^2} \frac{\operatorname{tr}\left( \mathbf{S}_t \bar{\bar{\mathbf{Q}}} \mathbf{S}_v \bar{\bar{\mathbf{Q}}} \right)}{(1+\delta_t)(1+\delta_v)} \right)_{t,v \in [T]} \right)^{-1} \in \mathbb{R}^{T,T}$$

$$\forall \mathbf{U} \in \mathbb{R}^{Td \times Td} : \quad \bar{\bar{\mathbf{Q}}}_2(\mathbf{U}) \equiv \bar{\bar{\mathbf{Q}}} \mathbf{U} \bar{\bar{\mathbf{Q}}} + \frac{1}{(Td)^2} \sum_{t,v=1}^T \frac{\operatorname{tr}(\mathbf{S}_t \bar{\bar{\mathbf{Q}}} \mathbf{U} \bar{\bar{\mathbf{Q}}}) P_{t,v} \bar{\bar{\mathbf{Q}}} \mathbf{S}_v \bar{\bar{\mathbf{Q}}}}{(1+\delta_t)(1+\delta_v)},$$

the result of Theorem 3 rewrites:

$$\left\| \mathbb{E}\left[ \tilde{\mathbf{Q}} \mathbf{U} \tilde{\mathbf{Q}} \right] - \bar{\bar{\mathbf{Q}}}_2(\mathbf{U}) \right\| \le O\left( \frac{1}{\sqrt{d}} \right) \qquad (9)$$

*Proof.* Given $i \in [n]$, let us note $\mathbf{Z}_{-i} = (\mathbf{z}_1, \ldots, \mathbf{z}_{i-1}, 0, \mathbf{z}_{i+1}, \ldots, \mathbf{z}_n)$ and $\tilde{\mathbf{Q}}_{-i} = (\frac{1}{Td}\mathbf{Z}_{-i}\mathbf{Z}_{-i}^\top + \mathbf{I}_{Td})^{-1}$, then we have the identity:

$$\tilde{\mathbf{Q}} - \tilde{\mathbf{Q}}_{-i} = \frac{1}{Td}\tilde{\mathbf{Q}}\mathbf{z}_i\mathbf{z}_i^\top\tilde{\mathbf{Q}}_{-i} \qquad \text{and} \qquad \tilde{\mathbf{Q}}\mathbf{z}_i = \frac{\tilde{\mathbf{Q}}_{-i}\mathbf{z}_i}{1 + \frac{1}{Td}\mathbf{z}_i^\top\tilde{\mathbf{Q}}_{-i}\mathbf{z}_i}. \qquad (10)$$

Given $\mathbf{u}, \mathbf{v} \in \mathbb{R}^{Td}$, such that $\|\mathbf{u}\|, \|\mathbf{v}\| \leq 1$, let us express:

$$\mathbb{E}\left[\frac{1}{Td}\mathbf{u}^\top\left(\tilde{\mathbf{Q}} - \bar{\tilde{\mathbf{Q}}}\right)\mathbf{U}\tilde{\mathbf{Q}}\mathbf{v}\right] = \frac{1}{n}\sum_{i=1}^n\mathbb{E}\left[\mathbf{u}^\top\tilde{\mathbf{Q}}\left(\frac{\mathbf{S}_i}{1+\boldsymbol{\delta}_i} - \mathbf{z}_i\mathbf{z}_i^\top\right)\bar{\tilde{\mathbf{Q}}}\mathbf{U}\tilde{\mathbf{Q}}\mathbf{v}\right] \qquad (11)$$

$$(12)$$

First, given $i \in [n]$, let us estimate thanks to (10):

$$\mathbb{E}\left[\mathbf{u}^\top\tilde{\mathbf{Q}}\mathbf{S}_i\bar{\tilde{\mathbf{Q}}}\mathbf{U}\tilde{\mathbf{Q}}\mathbf{v}\right] = \mathbb{E}\left[\mathbf{u}^\top\tilde{\mathbf{Q}}_{-i}\mathbf{S}_i\bar{\tilde{\mathbf{Q}}}\mathbf{U}\tilde{\mathbf{Q}}\mathbf{v}\right] - \frac{1}{Td}\mathbb{E}\left[\mathbf{u}^\top\tilde{\mathbf{Q}}\mathbf{z}_i\mathbf{z}_i^\top\tilde{\mathbf{Q}}_{-i}\mathbf{S}_i\bar{\tilde{\mathbf{Q}}}\mathbf{U}\tilde{\mathbf{Q}}\mathbf{v}\right]$$

Hölder inequality combined with Lemma 2 allows us to bound:

$$\frac{1}{Td}\left|\mathbb{E}\left[\mathbf{u}^\top\tilde{\mathbf{Q}}\mathbf{z}_i\mathbf{z}_i^\top\tilde{\mathbf{Q}}_{-i}\mathbf{S}_i\bar{\tilde{\mathbf{Q}}}\mathbf{U}\tilde{\mathbf{Q}}\mathbf{v}\right]\right| \leq \frac{1}{Td}\mathbb{E}\left[\left|\mathbf{u}^\top\tilde{\mathbf{Q}}\mathbf{z}_i\right|^2\right]^{\frac{1}{2}}\mathbb{E}\left[\left|\mathbf{z}_i^\top\tilde{\mathbf{Q}}_{-i}\mathbf{S}_i\bar{\tilde{\mathbf{Q}}}\mathbf{U}\tilde{\mathbf{Q}}\mathbf{v}\right|^2\right]^{\frac{1}{2}} \leq O\left(\frac{1}{d}\right),$$

one can thus deduce:

$$\mathbb{E}\left[\mathbf{u}^\top\tilde{\mathbf{Q}}\mathbf{S}_i\bar{\tilde{\mathbf{Q}}}\mathbf{U}\tilde{\mathbf{Q}}\mathbf{v}\right] = \mathbb{E}\left[\mathbf{u}^\top\tilde{\mathbf{Q}}_{-i}\mathbf{S}_i\bar{\tilde{\mathbf{Q}}}\mathbf{U}\tilde{\mathbf{Q}}\mathbf{v}\right] + O\left(\frac{1}{d}\right) = \mathbb{E}\left[\mathbf{u}^\top\tilde{\mathbf{Q}}_{-i}\mathbf{S}_i\bar{\tilde{\mathbf{Q}}}\mathbf{U}\tilde{\mathbf{Q}}_{-i}\mathbf{v}\right] + O\left(\frac{1}{d}\right). \qquad (13)$$

Second, one can also estimate thanks to Lemma 10:

$$\mathbb{E}\left[\mathbf{u}^\top\tilde{\mathbf{Q}}\mathbf{z}_i\mathbf{z}_i^\top\bar{\tilde{\mathbf{Q}}}\mathbf{U}\tilde{\mathbf{Q}}\mathbf{v}\right] = \mathbb{E}\left[\frac{\mathbf{u}^\top\tilde{\mathbf{Q}}_{-i}\mathbf{z}_i\mathbf{z}_i^\top\bar{\tilde{\mathbf{Q}}}\mathbf{U}\tilde{\mathbf{Q}}\mathbf{v}}{1 + \frac{1}{Td}\mathbf{z}_i^\top\mathbf{Q}_{-i}\mathbf{z}_i}\right] = \mathbb{E}\left[\frac{\mathbf{u}^\top\tilde{\mathbf{Q}}_{-i}\mathbf{z}_i\mathbf{z}_i^\top\bar{\tilde{\mathbf{Q}}}\mathbf{U}\tilde{\mathbf{Q}}\mathbf{v}}{1 + \delta_i}\right] + O\left(\frac{1}{\sqrt{d}}\right),$$

again thanks to Hölder inequality combined with Lemma 2 that allow us to bound:

$$\mathbb{E}\left[\left|\frac{\delta_i - \frac{1}{Td}\mathbf{z}_i^\top\mathbf{Q}_{-i}\mathbf{z}_i}{(1+\delta_i)\left(1 + \frac{1}{Td}\mathbf{z}_i^\top\mathbf{Q}_{-i}\mathbf{z}_i\right)}\right||\mathbf{u}^\top\tilde{\mathbf{Q}}_{-i}\mathbf{z}_i\mathbf{z}_i^\top\bar{\tilde{\mathbf{Q}}}\mathbf{U}\tilde{\mathbf{Q}}\mathbf{v}|\right]$$

$$\leq \mathbb{E}\left[\left|\delta_i - \frac{1}{Td}\mathbf{z}_i^\top\mathbf{Q}_{-i}\mathbf{z}_i\right|^2\right]^{\frac{1}{2}}\mathbb{E}\left[|\mathbf{u}^\top\tilde{\mathbf{Q}}_{-i}\mathbf{z}_i\mathbf{z}_i^\top\bar{\tilde{\mathbf{Q}}}\mathbf{U}\tilde{\mathbf{Q}}\mathbf{v}|^2\right]^{\frac{1}{2}} \leq O\left(\frac{1}{\sqrt{d}}\right),$$

The independence between $\mathbf{z}_i$ and $\tilde{\mathbf{Q}}_{-i}$ (and $\bar{\tilde{\mathbf{Q}}}$) then allow us to deduce (again with formula (10)):

$$\mathbb{E}\left[\mathbf{u}^\top\tilde{\mathbf{Q}}\mathbf{z}_i\mathbf{z}_i^\top\bar{\tilde{\mathbf{Q}}}\mathbf{U}\tilde{\mathbf{Q}}\mathbf{v}\right] = \mathbb{E}\left[\frac{\mathbf{u}^\top\tilde{\mathbf{Q}}_{-i}\mathbf{S}_i\bar{\tilde{\mathbf{Q}}}\mathbf{U}\tilde{\mathbf{Q}}_{-i}\mathbf{v}}{1+\delta_i}\right] + \frac{1}{Td}\mathbb{E}\left[\frac{\mathbf{u}^\top\tilde{\mathbf{Q}}_{-i}\mathbf{z}_i\mathbf{z}_i^\top\bar{\tilde{\mathbf{Q}}}\mathbf{U}\tilde{\mathbf{Q}}\mathbf{z}_i\mathbf{z}_i^\top\tilde{\mathbf{Q}}_{-i}\mathbf{v}}{1+\delta_i}\right] + O\left(\frac{1}{\sqrt{d}}\right). \qquad (14)$$

Let us inject (13) and (14) in (11) to obtain (again with an application of Hölder inequality and Lemma 2 that we do not detail this time):

$$\mathbb{E}\left[\mathbf{u}^\top\tilde{\mathbf{Q}}\left(\frac{\mathbf{S}_i}{1+\boldsymbol{\delta}_i} - \mathbf{z}_i\mathbf{z}_i^\top\right)\bar{\tilde{\mathbf{Q}}}\mathbf{U}\tilde{\mathbf{Q}}\mathbf{v}\right] = \frac{1}{Td}\mathbb{E}\left[\frac{\mathbf{u}^\top\tilde{\mathbf{Q}}_{-i}\mathbf{z}_i\mathbf{z}_i^\top\bar{\tilde{\mathbf{Q}}}\mathbf{U}\tilde{\mathbf{Q}}\mathbf{z}_i\mathbf{z}_i^\top\tilde{\mathbf{Q}}_{-i}\mathbf{v}}{\left(1 + \frac{1}{n}\mathbf{z}_i^\top\tilde{\mathbf{Q}}_{-i}\mathbf{z}_i\right)^2}\right] + O\left(\frac{1}{\sqrt{d}}\right),$$

$$= \frac{1}{Td}\frac{\mathbb{E}\left[\mathbf{u}^\top\tilde{\mathbf{Q}}_{-i}\mathbf{S}_i\tilde{\mathbf{Q}}_{-i}\mathbf{v}\right]}{(1+\delta_i)^2}\text{tr}\left(\mathbf{S}_i\bar{\tilde{\mathbf{Q}}}\mathbf{U}\bar{\tilde{\mathbf{Q}}}\right) + O\left(\frac{1}{\sqrt{d}}\right),$$

Putting all the estimations together, one finally obtains:

$$\left\|\mathbb{E}\left[\tilde{\mathbf{Q}}\mathbf{U}\tilde{\mathbf{Q}}\right] - \mathbb{E}\left[\bar{\tilde{\mathbf{Q}}}\mathbf{U}\bar{\tilde{\mathbf{Q}}}\right] - \frac{1}{(Td)^2}\sum_{i=1}^n\frac{\text{tr}\left(\mathbf{S}_i\bar{\tilde{\mathbf{Q}}}\mathbf{U}\bar{\tilde{\mathbf{Q}}}\right)}{(1+\delta_i)^2}\mathbb{E}\left[\tilde{\mathbf{Q}}_{-i}\mathbf{S}_i\tilde{\mathbf{Q}}_{-i}\right]\right\| \leq O\left(\frac{1}{\sqrt{d}}\right) \qquad (15)$$

One then see that if we introduce for any $\mathbf{V} \in \mathbb{R}^{n\times n}$ the block matrices:

- $\theta = \frac{1}{Td}\big(\frac{\mathbb{E}\big[\mathrm{tr}(\mathbf{S}_j\tilde{\mathbf{Q}}\mathbf{S}_i\tilde{\mathbf{Q}}^Y)\big]}{(1+\delta_i)(1+\delta_j)}\big)_{i,j\in[n]} \in \mathbb{R}^{n\times n}$

- $\theta(\mathbf{V}) = \frac{1}{Td}\big(\frac{\mathbb{E}\big[\mathrm{tr}(\mathbf{V}\tilde{\mathbf{Q}}\mathbf{S}_i\tilde{\mathbf{Q}}^Y)\big]}{1+\delta_i}\big)_{i\in[n]} \in \mathbb{R}^n,$

- $\theta(\mathbf{U},\mathbf{V}) = \frac{1}{Td}\mathbb{E}\Big[\mathrm{tr}(\mathbf{V}\tilde{\mathbf{Q}}\mathbf{U}\tilde{\mathbf{Q}}^Y)\Big] \in \mathbb{R},$

then, if $\|\mathbf{V}\| \le O(1)$, multiplying (15) with $\mathbf{V}$ and taking the trace leads to:

$$\theta(\mathbf{U},\mathbf{V}) = \Psi(\mathbf{U},\mathbf{V}) + \frac{1}{Td}\Psi(\mathbf{U})^\top\theta(\mathbf{V}) + O\left(\frac{1}{\sqrt{d}}\right), \tag{16}$$

Now, taking $\mathbf{U} = \frac{\mathbf{S}_1}{1+\delta_1},\dots,\frac{\mathbf{S}_n}{1+\delta_n}$, one gets the vectorial equation:

$$\theta(\mathbf{V}) = \Psi(\mathbf{V}) + \frac{1}{Td}\Psi\theta(\mathbf{V}) + O\left(\frac{1}{\sqrt{d}}\right),$$

When $(I_{Td} - \frac{1}{Td}\Psi)$ is invertible, one gets $\theta(\mathbf{V}) = (I_{Td} - \frac{1}{Td}\Psi)^{-1}\Psi(\mathbf{V}) + O\left(\frac{1}{\sqrt{d}}\right)$, and combining with (16), one finally obtains:

$$\theta(\mathbf{U},\mathbf{V}) = \Psi(\mathbf{U},\mathbf{V}) + \frac{1}{Td}\Psi(\mathbf{U})^\top(I_{Td} - \frac{1}{Td}\Psi)^{-1}\Psi(\mathbf{V}) + O\left(\frac{1}{\sqrt{d}}\right).$$

$\square$

## C.4 Estimation of the deterministic equivalent of $\mathbf{Q}^2$

**Theorem 4.** *Under the setting of Theorem 3, one can estimate:*

$$\big\|\mathbb{E}\big[\mathbf{Q}^2\big] - \mathbf{I}_n + \mathcal{D}_v\big\| \le O\left(\frac{1}{\sqrt{d}}\right), \tag{17}$$

*with, $\forall i \in [n]$:*

$$v_i \equiv \frac{1}{Td}\frac{\mathrm{tr}\left(\mathbf{S}_i\bar{\tilde{\mathbf{Q}}}\right)}{(1+\delta_i)^2} + \frac{1}{Td}\frac{\mathrm{tr}\left(\mathbf{S}_i\bar{\tilde{\mathbf{Q}}}_2(\mathbf{I}_n)\right)}{(1+\delta_i)^2}$$

*Proof.* The justifications are generally the same as in the proof of Theorem 3, we will thus allow ourselves to be quicker in this proof.

Using the definition of $\mathbf{Q} = \left(\frac{\mathbf{Z}^\top\mathbf{A}\mathbf{Z}}{Td} + \mathbf{I}_n\right)^{-1}$, we have that

$$\frac{\mathbf{Z}^\top\mathbf{Z}}{Td}\mathbf{Q} = \left(\frac{\mathbf{Z}^\top\mathbf{Z}}{Td} + \mathbf{I}_n - \mathbf{I}_n\right)\left(\frac{\mathbf{Z}^\top\mathbf{Z}}{Td} + \mathbf{I}_n\right)^{-1} = \mathbf{I}_n - \mathbf{Q} \tag{18}$$

and one can then let appear $\tilde{\mathbf{Q}}$ thanks to the relation:

$$\mathbf{Z}\mathbf{Q} = \tilde{\mathbf{Q}}\mathbf{Z}, \tag{19}$$

that finally gives us:

$$\mathbf{Q} = \mathbf{I}_n - \frac{1}{Td}\mathbf{Z}^\top\mathbf{Z}\mathbf{Q} = \mathbf{I}_n - \frac{1}{Td}\mathbf{Z}^\top\tilde{\mathbf{Q}}\mathbf{Z}$$

One can then express:

$$\mathbf{Q}^2 = \mathbf{I}_n - \frac{2}{Td}\mathbf{Z}^\top\tilde{\mathbf{Q}}\mathbf{Z} + \frac{1}{(Td)^2}\mathbf{Z}^\top\tilde{\mathbf{Q}}\mathbf{Z}\mathbf{Z}^\top\tilde{\mathbf{Q}}\mathbf{Z}$$

$$= \mathbf{I}_n - \frac{1}{Td}\mathbf{Z}^\top\tilde{\mathbf{Q}}\mathbf{Z} - \frac{1}{Td}\mathbf{Z}^\top\tilde{\mathbf{Q}}^2\mathbf{Z}.$$

Given $i, j \in [n]$, $i \neq j$, let us first estimate (thanks to Hölder inequality and Lemma 2):

$$\frac{1}{Td}\mathbb{E}\left[\mathbf{z}_i^\top \tilde{\mathbf{Q}}\mathbf{z}_j\right] = \frac{1}{Td}\frac{\mathbb{E}\left[\mathbf{z}_i^\top \tilde{\mathbf{Q}}_{-i,j}\mathbf{z}_j\right]}{(1+\delta_i)(1+\delta_j)} + O\left(\frac{1}{\sqrt{d}}\right) \leq O\left(\frac{1}{\sqrt{d}}\right),$$

since $\mathbb{E}[z_i] = \mathbb{E}[z_j] = 0$. Now, we consider the case $j = i$ to get:

$$\frac{1}{Td}\mathbb{E}\left[\mathbf{z}_i^\top \tilde{\mathbf{Q}}\mathbf{z}_i\right] = \frac{1}{Td}\frac{\mathbb{E}\left[\mathbf{z}_i^\top \tilde{\mathbf{Q}}_{-i}\mathbf{z}_i\right]}{(1+\delta_i)^2} + O\left(\frac{1}{\sqrt{d}}\right) = \frac{1}{Td}\frac{\operatorname{tr}\left(\mathbf{S}_i\bar{\tilde{\mathbf{Q}}}\right)}{(1+\delta_i)^2} + O\left(\frac{1}{\sqrt{d}}\right).$$

As before, we know that $\frac{1}{Td}\mathbb{E}\left[\mathbf{z}_i^\top \tilde{\mathbf{Q}}\mathbf{z}_j\right] \leq O\left(\frac{1}{\sqrt{d}}\right)$ if $i \neq j$. Considering $i \in [n]$, we thus are left to estimate:

$$\frac{1}{Td}\mathbb{E}\left[\mathbf{z}_i^\top \tilde{\mathbf{Q}}^2\mathbf{z}_j\right] = \frac{1}{Td}\frac{\operatorname{tr}\left(\mathbf{S}_i\bar{\tilde{\mathbf{Q}}}_2(\mathbf{I}_n)\right)}{(1+\delta_i)^2} + O\left(\frac{1}{\sqrt{d}}\right)$$

$\square$

## D  Risk Estimation (Proof of Theorem 1)

### D.1  Test Risk

The expected value of the MSE of the test data $\mathbf{x} \in \mathbb{R}^{T \times Td}$ concatenating the feature vector of all the tasks with the corresponding response variable $\mathbf{y} \in \mathbb{R}^{T \times Tq}$ reads as

$$\begin{aligned}
\mathcal{R}_{test}^\infty &= \frac{1}{T}\mathbb{E}[\|\mathbf{y} - g(\mathbf{x})\|_2^2] \\
&= \frac{1}{T}\mathbb{E}\left[\|\frac{\mathbf{x}^\top \mathbf{W}}{\sqrt{Td}} + \varepsilon - \frac{\mathbf{x}^\top \mathbf{AZQY}}{Td}\|_2^2\right] \\
&= \frac{1}{T}\mathbb{E}\left[\|\frac{\mathbf{x}^\top \mathbf{W}}{\sqrt{Td}} + \varepsilon - \frac{\mathbf{x}^\top \mathbf{AZQ}(\frac{\mathbf{Z}^\top \mathbf{W}}{\sqrt{Td}} + \varepsilon)}{Td}\|_2^2\right] \\
&= \frac{1}{T}\mathbb{E}\left[\|\frac{\mathbf{x}^\top \mathbf{W}}{\sqrt{Td}} + \varepsilon - \frac{\mathbf{x}^\top \mathbf{AZQZ}^\top \mathbf{W}}{Td\sqrt{Td}} - \frac{\mathbf{x}^\top \mathbf{AZQ}\varepsilon}{Td}\|_2^2\right] \\
&= \frac{1}{T}\mathbb{E}\left[\frac{\operatorname{tr}\left(\mathbf{W}^\top \boldsymbol{\Sigma}\mathbf{W}\right)}{Td} - \frac{2\operatorname{tr}\left(\mathbf{W}^\top \boldsymbol{\Sigma}\mathbf{AZQZ}^\top \mathbf{W}\right)}{(Td)^2} + \operatorname{tr}\left(\varepsilon^\top \varepsilon\right) + \frac{\operatorname{tr}\left(\mathbf{W}^\top \mathbf{ZQZ}^\top \mathbf{A}\boldsymbol{\Sigma}\mathbf{AZQZ}^\top \mathbf{W}\right)}{(Td)^3} + \right. \\
&\quad \left. \frac{\operatorname{tr}\left(\varepsilon^\top \mathbf{QZ}^\top \mathbf{A}\boldsymbol{\Sigma}\mathbf{AZQ}\varepsilon\right)}{(Td)^2}\right] \\
&= \frac{1}{T}\mathbb{E}\left[\frac{\operatorname{tr}\left(\mathbf{W}^\top \boldsymbol{\Sigma}\mathbf{W}\right)}{Td} - \frac{2\operatorname{tr}\left(\mathbf{W}^\top \boldsymbol{\Sigma}\mathbf{A}^{\frac{1}{2}}(\mathbf{I}_{Td} - \tilde{\mathbf{Q}})\mathbf{A}^{-\frac{1}{2}}\mathbf{W}\right)}{Td} + \right. \\
&\quad \left. \operatorname{tr}\left(\varepsilon^\top \varepsilon\right) + \frac{\operatorname{tr}\left(\mathbf{W}^\top \mathbf{ZQZ}^\top \mathbf{A}\boldsymbol{\Sigma}\mathbf{AZQZ}^\top \mathbf{W}\right)}{(Td)^3} + \frac{\operatorname{tr}\left(\varepsilon^\top \mathbf{QZ}^\top \mathbf{A}\boldsymbol{\Sigma}\mathbf{AZQ}\varepsilon\right)}{(Td)^2}\right] \\
&= \frac{\operatorname{tr}\left(\mathbf{W}^\top \boldsymbol{\Sigma}\mathbf{W}\right)}{Td} - 2\frac{\operatorname{tr}\left(\mathbf{W}^\top \boldsymbol{\Sigma}\mathbf{A}^{\frac{1}{2}}(\mathbf{I}_{Td} - \tilde{\mathbf{Q}})\mathbf{A}^{-\frac{1}{2}}\mathbf{W}\right)}{Td} + \operatorname{tr}\left(\varepsilon^\top \varepsilon\right) + \frac{\operatorname{tr}\left(\mathbf{W}^\top \boldsymbol{\Sigma}\mathbf{W}\right)}{Td} \\
&\quad - 2\frac{\operatorname{tr}\left(\mathbf{W}^\top \boldsymbol{\Sigma}A^{\frac{1}{2}}\bar{\tilde{\mathbf{Q}}}A^{-\frac{1}{2}}\mathbf{W}\right)}{Td} + \frac{\mathbf{W}^\top \mathbf{A}^{-\frac{1}{2}}\bar{\tilde{\mathbf{Q}}}_2(\mathbf{A})\mathbf{A}^{-\frac{1}{2}}\mathbf{W}}{Td} + \frac{1}{Td}\operatorname{tr}(\boldsymbol{\Sigma}_N\bar{\mathbf{Q}}_2) + O\left(\frac{1}{\sqrt{d}}\right)
\end{aligned}$$

The test risk can be further simplified as

$$\mathcal{R}_{test}^\infty = \operatorname{tr}\left(\boldsymbol{\Sigma}_N\right) + \frac{\mathbf{W}^\top \mathbf{A}^{-\frac{1}{2}}\bar{\tilde{\mathbf{Q}}}_2(\mathbf{A})\mathbf{A}^{-\frac{1}{2}}\mathbf{W}}{Td} + \frac{\operatorname{tr}\left(\boldsymbol{\Sigma}_N\bar{\mathbf{Q}}_2\right)}{Td} + O\left(\frac{1}{\sqrt{d}}\right)$$

## D.2 Train Risk

In this section, we derive the asymptotic risk for the training data.

**Theorem 5** (Asymptotic training risk). *Assuming that the training data vectors $\mathbf{x}_i^{(t)}$ and the test data vectors $\mathbf{x}^{(t)}$ are concentrated random vectors, and given the growth rate assumption (Assumption 2), it follows that:*

$$\mathcal{R}_{train}^{\infty} \leftrightarrow \frac{1}{Tn}\operatorname{tr}\left(\mathbf{W}^{\top}\mathbf{A}^{-1/2}\bar{\tilde{\mathbf{Q}}}\mathbf{A}^{-1/2}\mathbf{W}\right) - \frac{1}{Tn}\operatorname{tr}\left(\mathbf{W}^{\top}\mathbf{A}^{-1/2}\bar{\tilde{\mathbf{Q}}}_2(\mathbf{I}_{Td})\mathbf{A}^{-1/2}\mathbf{W}\right) + \frac{1}{Tn}\operatorname{tr}\left(\mathbf{\Sigma}_N\bar{\mathbf{Q}}_2\right)$$

*Proof.* We aim in this setting of regression, to compute the asymptotic theoretical training risk given by:

$$\mathcal{R}_{train}^{\infty} = \frac{1}{Tn}\mathbb{E}\left[\left\|\mathbf{Y} - \frac{\mathbf{Z}^{\top}\mathbf{A}\mathbf{Z}}{Td}\mathbf{Q}\mathbf{Y}\right\|_2^2\right]$$

Using the definition of $\mathbf{Q} = \left(\frac{\mathbf{Z}^{\top}\mathbf{A}\mathbf{Z}}{Td} + \mathbf{I}_{Td}\right)^{-1}$, we have that

$$\frac{\mathbf{Z}^{\top}\mathbf{A}\mathbf{Z}}{Td}\mathbf{Q} = \left(\frac{\mathbf{Z}^{\top}\mathbf{A}\mathbf{Z}}{Td} + \mathbf{I}_{Td} - \mathbf{I}_{Td}\right)\left(\frac{\mathbf{Z}^{\top}\mathbf{A}\mathbf{Z}}{Td} + \mathbf{I}_{Td}\right)^{-1} = \mathbf{I}_{Td} - \mathbf{Q}$$

Plugging back into the expression of the training risk then leads to

$$\mathcal{R}_{train}^{\infty} = \frac{1}{Tn}\mathbb{E}\left[\operatorname{tr}\left(\mathbf{Y}^{\top}\mathbf{Q}^2\mathbf{Y}\right)\right]$$

Using the definition of the linear generative model and in particular $\mathbf{Y} = \frac{\mathbf{Z}^{\top}\mathbf{W}}{\sqrt{Td}} + \varepsilon$, we get

$$\mathcal{R}_{train}^{\infty} = \frac{1}{Tn}\mathbb{E}\left[\operatorname{tr}\left(\frac{1}{\sqrt{Td}}\mathbf{Z}^{\top}\mathbf{W} + \varepsilon\right)^{\top}\mathbf{Q}^2\left(\frac{1}{\sqrt{Td}}\mathbf{Z}^{\top}\mathbf{W} + \varepsilon\right)\right]$$

$$= \frac{1}{Tn}\frac{1}{Td}\mathbb{E}\left[\operatorname{tr}\left(\mathbf{W}^{\top}\mathbf{Z}\mathbf{Q}^2\mathbf{Z}^{\top}\mathbf{W}\right)\right] + \frac{1}{Tn}\mathbb{E}\left[\operatorname{tr}\left(\varepsilon^{\top}\mathbf{Q}^2\varepsilon\right)\right]$$

To simplify this expression, we will introduced the so-called "coresolvent" defined as:

$$\tilde{\mathbf{Q}} = \left(\frac{\mathbf{A}^{\frac{1}{2}}\mathbf{Z}\mathbf{Z}^{\top}\mathbf{A}^{\frac{1}{2}}}{Td} + \mathbf{I}_{Td}\right)^{-1},$$

Employing the elementary relation $\mathbf{A}^{\frac{1}{2}}\mathbf{Z}\mathbf{Q} = \tilde{\mathbf{Q}}\mathbf{A}^{\frac{1}{2}}\mathbf{Z}$, one obtains:

$$\frac{1}{Td}\mathbf{Z}\mathbf{Q}^2\mathbf{Z}^{\top} = \frac{1}{Td}\mathbf{A}^{-\frac{1}{2}}\tilde{\mathbf{Q}}\mathbf{A}^{\frac{1}{2}}\mathbf{Z}\mathbf{Q}\mathbf{Z}^{\top} = \mathbf{A}^{-\frac{1}{2}}\tilde{\mathbf{Q}}^2\frac{\mathbf{A}^{\frac{1}{2}}\mathbf{Z}\mathbf{Z}^{\top}\mathbf{A}^{\frac{1}{2}}}{Td}\mathbf{A}^{-\frac{1}{2}} = \mathbf{A}^{-\frac{1}{2}}\tilde{\mathbf{Q}}\mathbf{A}^{-\frac{1}{2}} - \mathbf{A}^{-\frac{1}{2}}\tilde{\mathbf{Q}}^2\mathbf{A}^{-\frac{1}{2}},$$

Therefore we further get

$$\mathcal{R}_{train}^{\infty} = \frac{1}{Tn}\mathbb{E}\left[\operatorname{tr}\left(\mathbf{W}^{\top}\mathbf{A}^{-1/2}\tilde{\mathbf{Q}}\mathbf{A}^{-1/2}\mathbf{W}\right)\right] - \frac{1}{Tn}\mathbb{E}\left[\operatorname{tr}\left(\mathbf{W}^{\top}\mathbf{A}^{-1/2}\tilde{\mathbf{Q}}^2\mathbf{A}^{-1/2}\mathbf{W}\right)\right] + \frac{1}{Tn}\mathbb{E}\left[\operatorname{tr}\left(\varepsilon^{\top}\mathbf{Q}^2\varepsilon\right)\right]$$

Using deterministic equivalents in Lemma 1, the training risk then leads to

$$\mathcal{R}_{train}^{\infty} = \frac{1}{Tn}\operatorname{tr}\left(\mathbf{W}^{\top}\mathbf{A}^{-1/2}\bar{\tilde{\mathbf{Q}}}\mathbf{A}^{-1/2}\mathbf{W}\right) - \frac{1}{Tn}\operatorname{tr}\left(\mathbf{W}^{\top}\mathbf{A}^{-1/2}\bar{\tilde{\mathbf{Q}}}_2(\mathbf{I}_{Td})\mathbf{A}^{-1/2}\mathbf{W}\right) + \frac{1}{Tn}\operatorname{tr}\left(\mathbf{\Sigma}_N\bar{\mathbf{Q}}_2\right) + O\left(\frac{1}{\sqrt{d}}\right)$$

$\square$

# E  Interpretation and insights of the theoretical analysis

## E.1  Analysis of the test risk

We recall the test risk as

$$\mathcal{R}_{test}^{\infty} = \operatorname{tr}\left(\mathbf{\Sigma}_N\right) + \frac{\mathbf{W}^{\top}\mathbf{A}^{-\frac{1}{2}}\bar{\tilde{\mathbf{Q}}}_2(\mathbf{A})\mathbf{A}^{-\frac{1}{2}}\mathbf{W}}{Td} + \frac{\operatorname{tr}\left(\mathbf{\Sigma}_N\bar{\mathbf{Q}}_2\right)}{Td} + O\left(\frac{1}{\sqrt{d}}\right)$$

The test risk is composed of a signal term of a signal term $\mathcal{S} = \frac{\mathbf{W}^{\top}\mathbf{A}^{-\frac{1}{2}}\bar{\tilde{\mathbf{Q}}}_2(\mathbf{A})\mathbf{A}^{-\frac{1}{2}}\mathbf{W}}{Td}$ and a noise term $\mathcal{N} = \frac{\operatorname{tr}\left(\mathbf{\Sigma}_N\bar{\mathbf{Q}}_2\right)}{Td}$.

## E.2 Interpretation of the signal term

Let's denote by $\bar{\boldsymbol{\Sigma}} = \sum_{t=1}^{T} \frac{n_t d_t}{Td(1+\delta_t)^2} \boldsymbol{\Sigma}^{(t)}$ and $\tilde{\boldsymbol{\Sigma}} = \sum_{t=1}^{T} \frac{c_0}{1+\delta_t} \boldsymbol{\Sigma}^{(t)}$. The signal term reads as

$$\mathcal{S} = \mathbf{W}^{\top} \mathbf{A}^{-\frac{1}{2}} \bar{\tilde{\mathbf{Q}}}_2(\mathbf{A}) \mathbf{A}^{-\frac{1}{2}} \mathbf{W}.$$

Using the following identity,

$$\mathbf{A}^{-\frac{1}{2}} \bar{\tilde{\mathbf{Q}}}_2(\mathbf{A}) \mathbf{A}^{-\frac{1}{2}} = \mathbf{A}^{-\frac{1}{2}} \bar{\mathbf{Q}} \mathbf{A}^{\frac{1}{2}} \left( \mathbf{I} + \bar{\boldsymbol{\Sigma}} \right) \mathbf{A}^{\frac{1}{2}} \bar{\mathbf{Q}} \mathbf{A}^{-\frac{1}{2}}$$

$$= \left( \mathbf{A}\tilde{\boldsymbol{\Sigma}} + \mathbf{I} \right)^{-1} \left( \mathbf{I} + \bar{\boldsymbol{\Sigma}} \right) \left( \mathbf{A}\tilde{\boldsymbol{\Sigma}} + \mathbf{I} \right)^{-1}$$

This finally leads to

$$\mathcal{S} = \mathbf{W}^{\top} \left( \mathbf{A}\tilde{\boldsymbol{\Sigma}} + \mathbf{I} \right)^{-1} \left( \mathbf{I} + \bar{\boldsymbol{\Sigma}} \right) \left( \mathbf{A}\tilde{\boldsymbol{\Sigma}} + \mathbf{I} \right)^{-1} \mathbf{W}$$

The matrix $\mathcal{H} = \left( \mathbf{A}\tilde{\boldsymbol{\Sigma}} + \mathbf{I} \right)^{-1} \left( \mathbf{I} + \bar{\boldsymbol{\Sigma}} \right) \left( \mathbf{A}\tilde{\boldsymbol{\Sigma}} + \mathbf{I} \right)^{-1}$ is responsible to amplifying the signal $\mathbf{W}^{\top}\mathbf{W}$ in order to let the test risk to decrease more or less. It is is decreasing as function of the number of samples in the tasks $n_t$. Furthermore it is composed of two terms (from the independent training $\mathbf{W}_t^{\top}\mathbf{W}$) and the cross term $\mathbf{W}_t^{\top}\mathbf{W}_v$ for $t \neq v$. Both terms decreases as function of the number of samples $n_t$, smaller values of $\gamma_t$ and increasing value of $\lambda$. The cross term depends on the matrix $\boldsymbol{\Sigma}_t^{-1}\boldsymbol{\Sigma}_v$ which materializes the covariate shift between the tasks. More specifically, if the features are aligned $\boldsymbol{\Sigma}_t^{-1}\boldsymbol{\Sigma}_v = I$ and the cross term is maximal while for bigger Fisher distance between the covariance of the tasks, the correlation is not favorable for multi task learning. To be more specific the off-diagonal term of $\mathcal{H}$ are responsible for the cross term therefore for the multi tasks and the diagonal elements are responsible for the independent terms.

To analyze more the element of $\mathcal{H}$, let's consider the case where $\boldsymbol{\Sigma}^{(t)} = \mathbf{I}$ and $\gamma_t = \gamma$. In this case the diagonal and non diagonal elements $\mathbf{D}_{IL}$ and $\mathbf{C}_{MTL}$ are respectively given by

$$\mathbf{D}_{IL} = \frac{(c_0(\lambda + \gamma) + 1)^2 + c_0^2\lambda^2}{(c_0(\lambda + \gamma) + 1)^2 - c_0^2\lambda^2}, \quad \mathbf{C}_{MTL} = \frac{-2c_0\lambda(c_0(\lambda + \gamma) + 1)}{(c_0(\lambda + \gamma) + 1)^2 - c_0^2\lambda^2}$$

Both function are decreasing function of $\lambda$, $1/\gamma$ and $c_0$.

## E.3 Interpretation and insights of the noise terms

We recall the definition of the noise term $\mathcal{N}$ as

$$\mathcal{N} = \text{tr} \left( \boldsymbol{\Sigma}_N \left( \mathbf{A}^{-1} + \boldsymbol{\Sigma} \right)^{-1} \right)$$

Now at the difference of the signal term there are no cross terms due to the independence between the noise of the different tasks. In this case on the diagonal elements of $\left( \mathbf{A}^{-1} + \boldsymbol{\Sigma} \right)^{-1}$ matters. This diagonal term is increasing for an increasing value of the sample size, the value of $\lambda$. Therefore this term is responsible for the negative transfer. In the specific case where $\boldsymbol{\Sigma}^{(t)} = \mathbf{I}_d$ and $\gamma_t = \gamma$ for all task $t$, the diagonal terms read as

$$\mathbf{N}_{NT} = \frac{(c_0(\lambda + \gamma)^2 + (\lambda + \gamma) - c_0\lambda^2)^2 + \lambda^2}{\left( (c_0(\lambda + \gamma) + 1)^2 - c_0^2\lambda^2 \right)^2}$$

## E.4 Optimal Lambda

The test risk in the particular of identity covariance matrix can be rewritten as

$$\mathcal{R}_{test}^{\infty} = \mathbf{D}_{IL} \left( \|\mathbf{W}_1\|_2^2 + \|\mathbf{W}_2\|_2^2 \right) + \mathbf{C}_{MTL}\mathbf{W}_1^{\top}\mathbf{W}_2 + \mathbf{N}_{NT}\text{tr}\boldsymbol{\Sigma}_n.$$

Deriving $\mathcal{R}_{test}^{\infty}$ with respect to $\lambda$ leads after some algebraic calculus to

$$\lambda^{\star} = \frac{n}{d}SNR - \frac{\gamma}{2}$$

where the signal noise ratio is composed of the independent signal to noise ratio and the cross signal to noise ratio $SNR = \frac{\|\mathbf{W}_1\|_2^2 + \mathbf{W}_2\|_2^2}{\text{tr}\boldsymbol{\Sigma}_n} + \frac{\mathbf{W}_1^{\top}\mathbf{W}_2}{\text{tr}\boldsymbol{\Sigma}_n}$

# F Theoretical Estimations

## F.1 Estimation of the training and test risk

The different theorems depends on the ground truth $\mathbf{W}$ that needs to be estimated through $\hat{\mathbf{W}}$.

To estimate the test risk, one needs to estimate functionals of the form $\mathbf{W}^\top \mathbf{M} \hat{\mathbf{W}}$ and $\varepsilon^\top \mathbf{M} \varepsilon$ for any matrix $\mathbf{M}$. Using the expression of $\mathbf{W} = \mathbf{AZQY}$, we start computing $\hat{\mathbf{W}}^\top \mathbf{M} \hat{\mathbf{W}}$

$$\hat{\mathbf{W}}^\top \mathbf{M} \mathbf{W} = \mathbf{Y}^\top \mathbf{Q} \mathbf{Z}^\top \mathbf{A} \mathbf{M} \mathbf{A} \mathbf{Z} \mathbf{Q} \mathbf{Y}$$

Using the generative model for $\mathbf{Y} = \frac{\mathbf{Z}^\top \mathbf{W}}{\sqrt{Td}} + \varepsilon$, we obtain

$$\mathbb{E}\left[\hat{\mathbf{W}}^\top \mathbf{M} \mathbf{W}\right] = \mathbb{E}\left[\left(\frac{\mathbf{Z}^\top \mathbf{W}}{\sqrt{Td}} + \varepsilon\right)^\top \mathbf{Q} \mathbf{Z}^\top \mathbf{A} \mathbf{M} \mathbf{A} \mathbf{Z} \mathbf{Q}\left(\frac{\mathbf{Z}^\top \mathbf{W}}{\sqrt{Td}} + \varepsilon\right)\right]$$

$$= \frac{1}{Td}\mathbb{E}\left[\mathbf{W}^\top \mathbf{Z} \mathbf{Q} \mathbf{Z}^\top \mathbf{A} \mathbf{M} \mathbf{A} \mathbf{Z} \mathbf{Q} \mathbf{Z}^\top \mathbf{W}\right] + \mathbb{E}\left[\varepsilon^\top \mathbf{Q} \mathbf{Z}^\top \mathbf{A} \mathbf{M} \mathbf{A} \mathbf{Z} \mathbf{Q} \varepsilon\right]$$

Employing the elementary relation $\mathbf{A}^{\frac{1}{2}} \mathbf{Z} \mathbf{Q} = \tilde{\mathbf{Q}} \mathbf{A}^{\frac{1}{2}} \mathbf{Z}$, one obtains:

$$\mathbb{E}\left[\hat{\mathbf{W}}^\top \mathbf{M} \mathbf{W}\right] = \frac{1}{Td}\mathbb{E}\left[\mathbf{W}^\top \mathbf{A}^{-\frac{1}{2}} \tilde{\mathbf{Q}} \mathbf{A}^{\frac{1}{2}} \mathbf{Z}^\top \mathbf{Z} \mathbf{A}^{\frac{1}{2}} \mathbf{A}^{\frac{1}{2}} \mathbf{M} \mathbf{A}^{\frac{1}{2}} \tilde{\mathbf{Q}} \mathbf{A}^{\frac{1}{2}} \mathbf{Z} \mathbf{Z}^\top \mathbf{W}\right] + \mathbb{E}\left[\varepsilon^\top \mathbf{Q} \mathbf{Z}^\top \mathbf{A} \mathbf{M} \mathbf{A} \mathbf{Z} \mathbf{Q} \varepsilon\right]$$

$$= \mathbb{E}\left[\mathbf{W}^\top \mathbf{A}^{-\frac{1}{2}}\left(\mathbf{I} - \tilde{\mathbf{Q}}\right) \mathbf{A}^{\frac{1}{2}} \mathbf{M} \mathbf{A}^{\frac{1}{2}}\left(\mathbf{I} - \tilde{\mathbf{Q}}\right) \mathbf{A}^{-\frac{1}{2}} \mathbf{W}\right] + \mathbb{E}\left[\varepsilon^\top \mathbf{Q} \mathbf{Z}^\top \mathbf{A} \mathbf{M} \mathbf{A} \mathbf{Z} \mathbf{Q} \varepsilon\right]$$

$$= \mathbb{E}\left[\mathbf{W}^\top \mathbf{M} \mathbf{W}\right] - 2\mathbb{E}\left[\mathbf{W}^\top \mathbf{M} \mathbf{A}^{\frac{1}{2}} \tilde{\mathbf{Q}} \mathbf{A}^{-\frac{1}{2}} \mathbf{W}\right] + \mathbb{E}\left[\mathbf{W}^\top \mathbf{A}^{-\frac{1}{2}} \tilde{\mathbf{Q}} \mathbf{A}^{\frac{1}{2}} \mathbf{M} \mathbf{A}^{\frac{1}{2}} \tilde{\mathbf{Q}} \mathbf{A}^{-\frac{1}{2}} \mathbf{W}\right]$$

$$+ \mathbb{E}\left[\varepsilon^\top \mathbf{Q} \mathbf{Z}^\top \mathbf{A} \mathbf{M} \mathbf{A} \mathbf{Z} \mathbf{Q} \varepsilon\right]$$

Using the deterministic equivalent of Lemma 1, we obtain

$$\hat{\mathbf{W}}^\top \mathbf{M} \hat{\mathbf{W}} \leftrightarrow \mathbf{W}^\top \mathbf{M} \mathbf{W} - 2\mathbf{W}^\top \mathbf{A}^{-\frac{1}{2}} \bar{\tilde{\mathbf{Q}}} \mathbf{A}^{\frac{1}{2}} \mathbf{M} \mathbf{W} + \mathrm{tr} \mathbf{\Sigma}_n \mathbf{M}(\mathbf{A}^{\frac{1}{2}} \mathbf{M} \mathbf{A}^{\frac{1}{2}}) + \mathbf{W}^\top \mathbf{A}^{-\frac{1}{2}} \bar{\tilde{\mathbf{Q}}}_2(\mathbf{A}^{\frac{1}{2}} \mathbf{M} \mathbf{A}^{\frac{1}{2}}) \mathbf{A}^{-\frac{1}{2}} \mathbf{W}$$

$$\leftrightarrow \mathbf{W}^\top\left(\mathbf{M} - 2\mathbf{A}^{-\frac{1}{2}} \bar{\tilde{\mathbf{Q}}} \mathbf{A}^{\frac{1}{2}} \mathbf{M} + \mathbf{A}^{-\frac{1}{2}} \bar{\tilde{\mathbf{Q}}}_2(\mathbf{A}^{\frac{1}{2}} \mathbf{M} \mathbf{A}^{\frac{1}{2}}) \mathbf{A}^{-\frac{1}{2}}\right) \mathbf{W} + \mathrm{tr} \mathbf{\Sigma}_n \bar{\tilde{\mathbf{Q}}}_2(\mathbf{A}^{\frac{1}{2}} \mathbf{M} \mathbf{A}^{\frac{1}{2}})$$

$$\leftrightarrow \mathbf{W}^\top \kappa(\mathbf{M}) \mathbf{W} + \mathrm{tr} \mathbf{\Sigma}_n \bar{\tilde{\mathbf{Q}}}_2(\mathbf{A}^{\frac{1}{2}} \mathbf{M} \mathbf{A}^{\frac{1}{2}})$$

where We define the mapping $\kappa : \mathbb{R}^{Td \times Td} \to \mathbb{R}^{q \times q}$ as follows

$$\kappa(\mathbf{M}) = \mathbf{M} - 2\mathbf{A}^{-\frac{1}{2}} \bar{\tilde{\mathbf{Q}}} \mathbf{A}^{\frac{1}{2}} \mathbf{M} + \mathbf{A}^{-\frac{1}{2}} \bar{\tilde{\mathbf{Q}}}_2(\mathbf{A}^{\frac{1}{2}} \mathbf{M} \mathbf{A}^{\frac{1}{2}}) \mathbf{A}^{-\frac{1}{2}}.$$

## F.2 Estimation of the noise covariance

The estimation of the noise covariance remains a technical challenge in this process. However, when the noise covariance is isotropic, it is sufficient to estimate only the noise variance. By observing that

$$\lim_{\lambda \to 0, \gamma \to \infty} \mathcal{R}_{train}^\infty = \sigma^2 \frac{\mathrm{tr} \mathbf{Q}_2}{kn},$$

we can estimate the noise level from the training risk evaluated at large $\gamma$ and $\lambda = 0$.

# G Multivariate Time Series Forecasting

A version of this work with a stronger focus on time series applications can be found at [20].

## G.1 Related Work

**Multivariate Time Series Forecasting.** Previous work has explored multivariate time series methods to enrich task representations and enhance diversity across time series applications [18].

Despite these advancements, random matrix theory, widely applied in multi-task learning, remains underutilized in multivariate time series contexts, particularly for multivariate time series forecasting (MTSF) [52, 54]. MTSF is common in applications like medical data [6], electricity consumption [49], temperatures [27], and stock prices [44]. Various methods, from classical tools [7, 45] and statistical approaches like ARIMA [3, 4] to deep learning techniques [8, 19, 32, 38, 40, 51, 53, 55, 56], have been developed for this task. Some studies prefer univariate models for multivariate forecasting [32], while others introduce channel-wise attention mechanisms [19]. We aim to enhance a univariate model by integrating a regularization technique from random matrix theory and multi-task learning.

## G.2 Architecture and Training Parameters

**Architectures without MTL regularization.** We follow Chen et al. [8], Nie et al. [32], and to ensure a fair comparison of baselines, we apply the reversible instance normalization (`RevIN`) of Kim et al. [21]. All the baselines presented here are univariate i.e. no operation is performed by the network along the channel dimension. For the `PatchTST` baseline [32], we used the official implementation than can be found on Github. The network used in `Transformer` follows the one in [19], using a single layer `Transformer` with one head of attention, while `RevIN` normalization and denormalization are applied respectively before and after the neural network function. The dimension of the model is $d_{\mathrm{m}} = 16$ and remains the same in all our experiments. `DLinearU` is a single linear layer applied for each channel to directly project the subsequence of historical length into the forecasted subsequence of prediction length. It is the univariate extension of the multivariate DLinear, DLinearM, used in Zeng et al. [53]. The implementation of `SAMformer` can be found here, and for the `iTransformer` architecture here. These two multivariate models serve as baseline comparisons. We reported the results found in [19] and [24]. For all of our experiments, we train our baselines `PatchTST`, `DLinearU` and `Transformer` with the Adam optimizer [22], a batch size of 32 for the `ETT` datasets and 256 for the `Weather` dataset, and the learning rates summarized in Table 2.

**Architectures with MTL Regularization.** We implemented the univariate `PatchTST`, `DLinearU`, and `Transformer` baselines with MTL regularization. Initially, we scale the inputs twice using `RevIN` normalization. The first scaling is applied to the univariate components, and the second scaling is applied to the multivariate components. For each channel, we then apply our model without MTL regularization. The outputs are concatenated along the channel dimension, and this concatenation is flattened to form a matrix of shape (batch size, $q \times T$), where $q$ is the prediction horizon and $T$ is the number of channels. We then learn a square matrix $W$ of shape $(q \times T) \times (q \times T)$ for projection and reshape the result to obtain an output of shape (batch size, $q$, $T$). This method can be applied on top of any univariate model. Our regularized loss has been introduced in 5.4.

**Training parameters.** The training/validation/test split is $12/4/4$ months on the `ETT` datasets and $70\%/20\%/10\%$ on the `Weather` dataset. We use a look-back window $d = 336$ for `PatchTST` and $d = 512$ for `DLinearU` and `Transformer`, using a sliding window with stride 1 to create the sequences. The training loss is the MSE. Training is performed during 100 epochs and we use early stopping with a patience of 5 epochs. For each dataset, baselines, and prediction horizon $H \in \{96, 192, 336, 720\}$, each experiment is run 3 times with different seeds, and we display the average of the test MSE over the 3 trials in Table 1.

Table 2: Learning rates used in our experiments.

| Dataset | ETTh1/ETTh2 | Weather |
|---|---|---|
| Learning rate | 0.001 | 0.0001 |

## G.3 Datasets

We conduct our experiments on 3 publicly available datasets of real-world time series, widely used for multivariate long-term forecasting [8, 32, 51]. The 2 Electricity Transformer Temperature datasets ETTh1, and ETTh2 [55] contain the time series collected by electricity transformers from July 2016 to July 2018. Whenever possible, we refer to this set of 2 datasets as ETT. Weather [27] contains the time series of meteorological information recorded by 21 weather indicators in 2020. It should be noted Weather is large-scale datasets. The ETT datasets can be downloaded here while the Weather

dataset can be downloaded here. Table 3 sums up the characteristics of the datasets used in our experiments.

Table 3: Characteristics of the multivariate time series datasets used in our experiments.

| Dataset | ETTh1/ETTh2 | Weather |
|---|---|---|
| # features | 7 | 21 |
| # time steps | 17420 | 52696 |
| Granularity | 1 hour | 10 minutes |

## G.4 Additional Experiments.

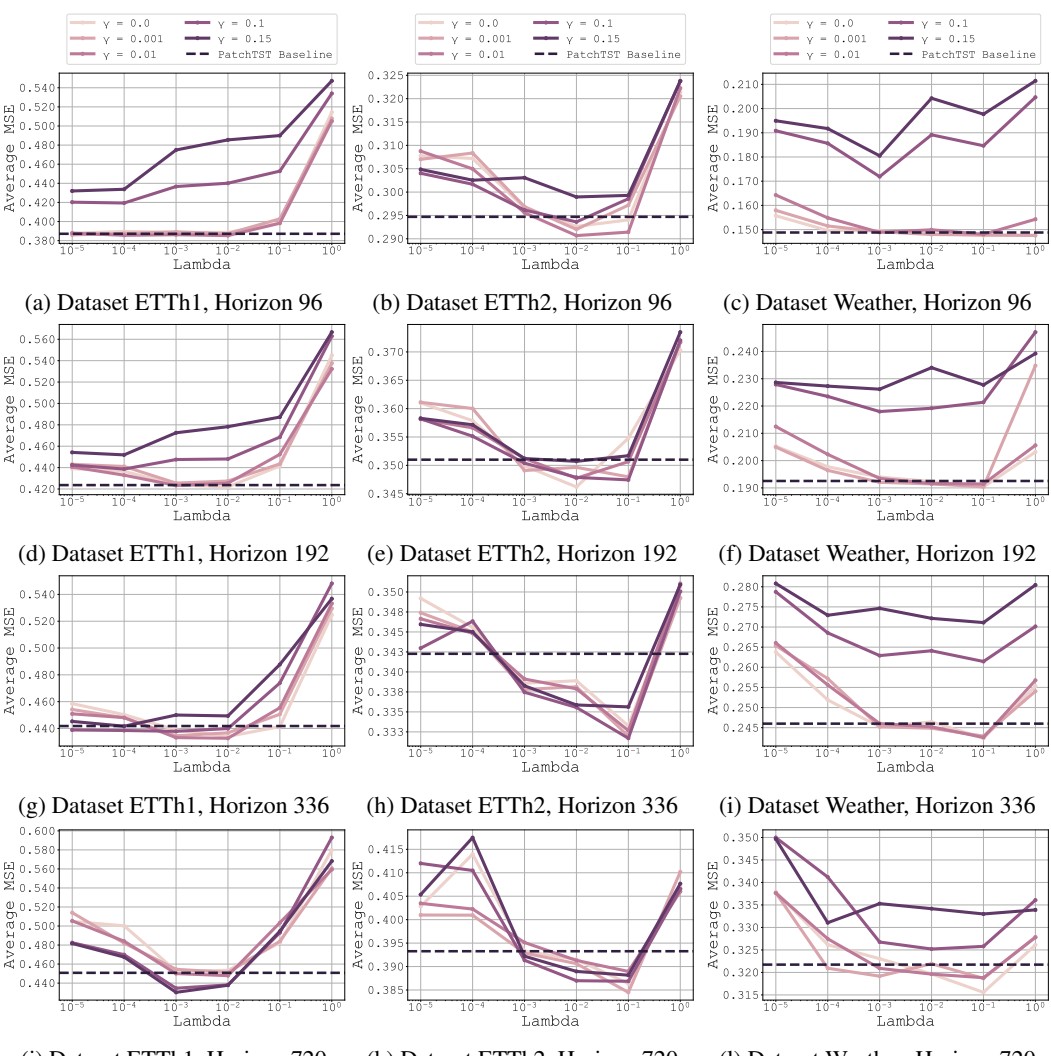

Figure 4: Results of our optimization method on different datasets and horizons averaged across 3 different seeds for each gamma and lambda values for the `PatchTST` baseline

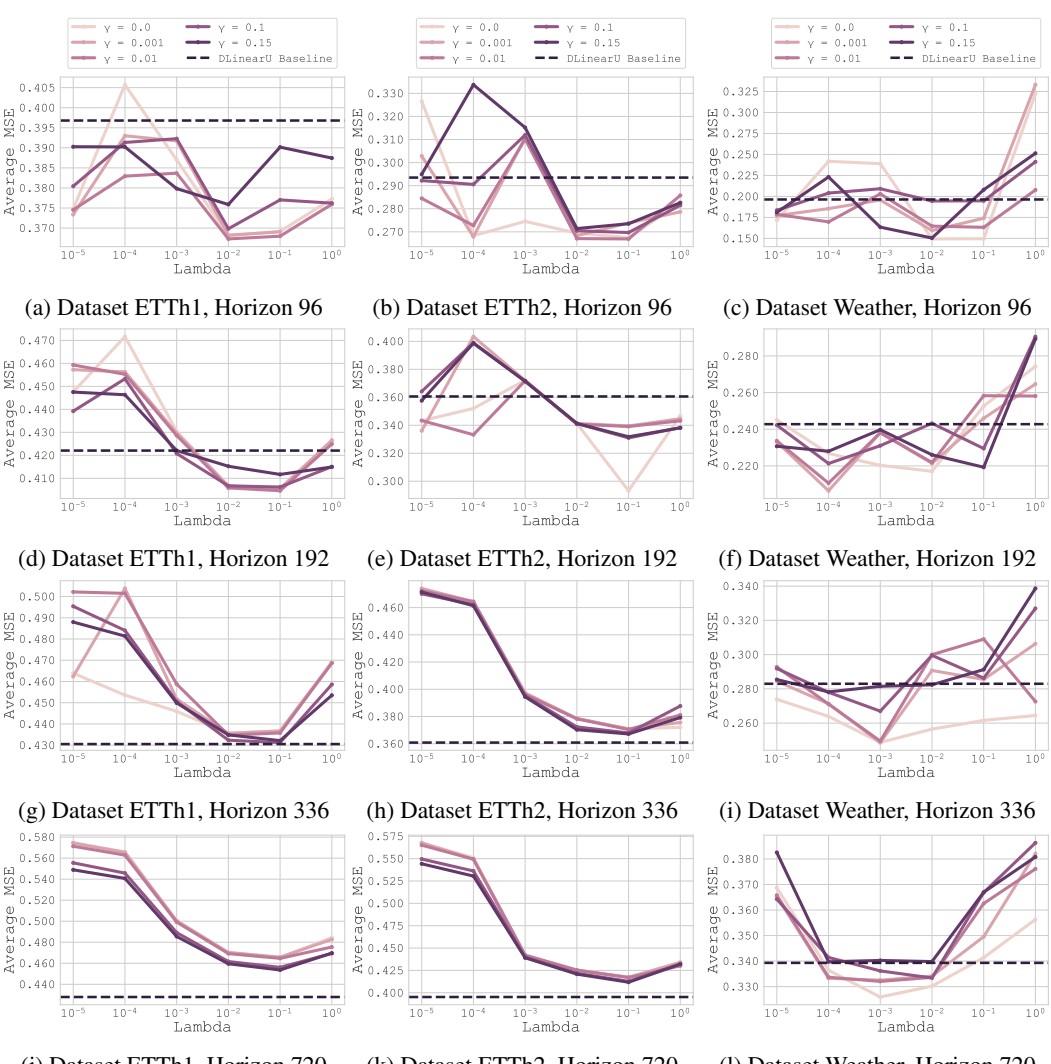

(a) Dataset ETTh1, Horizon 96     (b) Dataset ETTh2, Horizon 96     (c) Dataset Weather, Horizon 96

(d) Dataset ETTh1, Horizon 192     (e) Dataset ETTh2, Horizon 192     (f) Dataset Weather, Horizon 192

(g) Dataset ETTh1, Horizon 336     (h) Dataset ETTh2, Horizon 336     (i) Dataset Weather, Horizon 336

(j) Dataset ETTh1, Horizon 720     (k) Dataset ETTh2, Horizon 720     (l) Dataset Weather, Horizon 720

Figure 5: Results of our optimization method on different datasets and horizons averaged across 3 different seeds for each gamma and lambda values for the `DLinearU` baseline

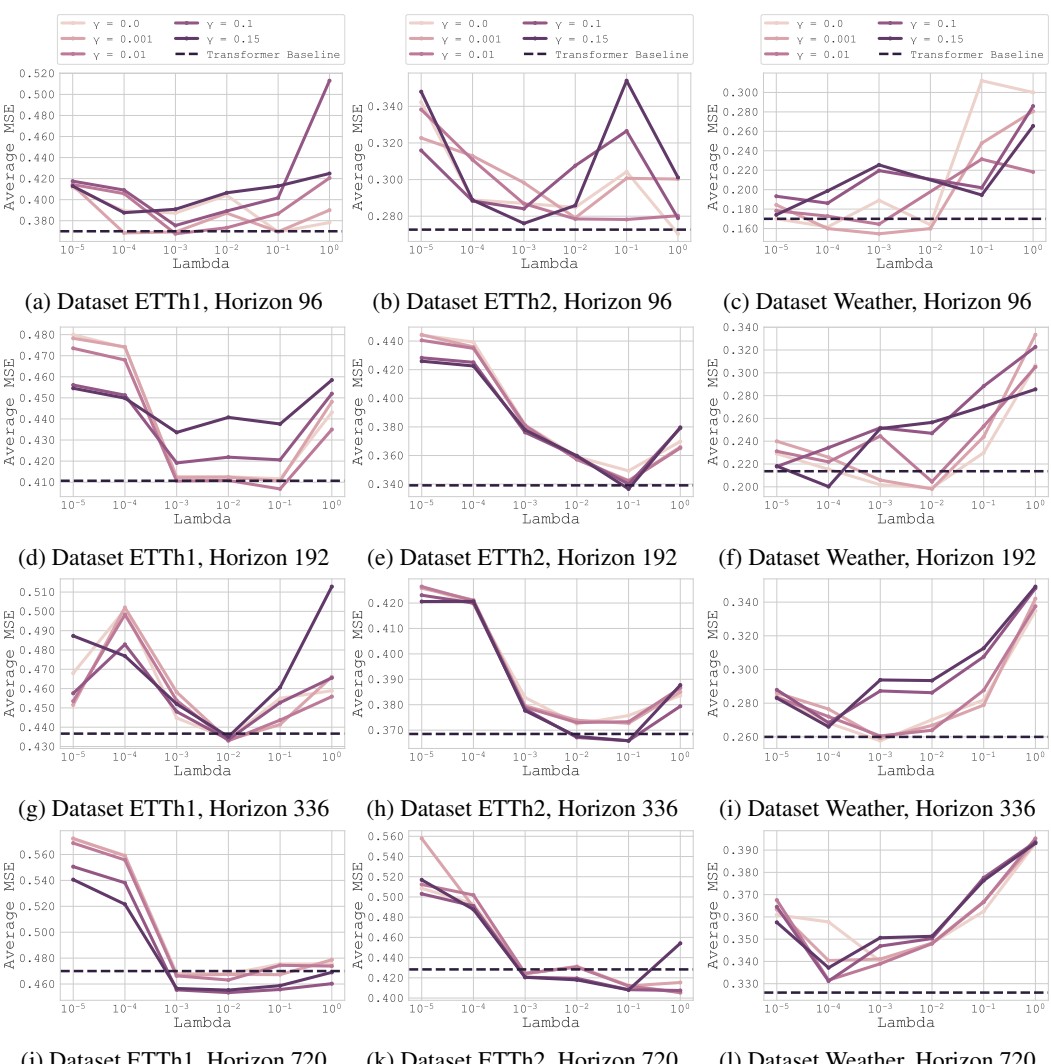

Figure 6: Results of our optimization method on different datasets and horizons averaged across 3 different seeds for each gamma and lambda values for the `Transformer` baseline

# H Limitations

While the study provides valuable insights through its theoretical analysis within a linear framework, it is important to acknowledge its limitations. The linear approach serves as a solid foundation for understanding more complex models, but its practical applications may be constrained. Linear models, though mathematically tractable and often easier to interpret, might not fully capture the intricacies and nonlinear relationships present in real-world data, especially in the context of multivariate time series forecasting.

To address this limitation, we decided to extend our algorithm's application to more complex models, specifically within the nonlinear setting of neural networks. This transition aims to evaluate whether the theoretical insights derived from the linear framework hold true empirically when applied to neural networks. As part of this endeavor, an optimal parameter lambda was selected by an oracle, leading to promising results, as detailed in Section 5.4. This oracle-based selection underscores the potential efficacy of our approach when appropriately tuned, even in more complex, nonlinear contexts.

It is important to note that the limitations are not related to the real-world data itself, as our setting performs well in the context of multi-task regression for real-world data, as shown in Section 5.2. The difficulty arises from transitioning from a linear to a nonlinear model. The results in Section 5.4 are particularly encouraging, demonstrating that our method can improve upon univariate baselines by regularizing with an optimal lambda, as indicated by our oracle. While the oracle provides an upper bound on performance, actual implementation would require robust methods for hyperparameter optimization in non-linear scenarios, which remains an open area for further research.

By expanding the scope of our theoretical framework to encompass nonlinear models, we pave the way for future work that could focus on the theoretical analysis of increasingly complex architectures

