# OpenReview forum: "Analysing Multi-Task Regression via Random Matrix Theory with Application to Time Series Forecasting"
_NeurIPS.cc/2024/Conference — NeurIPS 2024 spotlight_

### Official Review · Reviewer_EGU1 · 2024-07-09

**Soundness:** 3
**Presentation:** 3
**Contribution:** 3
**Rating:** 4
**Confidence:** 3

**Summary:**

This paper analyzes a linear multi-task regression framework. It derives formulas for asymptotic train and test risk. The formulas provide insights in how the raw data covariances, singal-generating hyperplanes, noise levels, and size of data sets affect the risk. Motivated by the analysis on the linear framework, experiments on multivariate time series forecasting are performed to investigate the effectiveness of MTL regularization.

**Strengths:**

- The paper is clearly written. It outlines main points and is easy to follow.
- Throughout analyses, which use random matrix theory, on the multi-task linear regression framework are performed. Specifically, the asymptotic train and test risks are derived for analyzing the behavior of the framework.
- The analyses provide useful insights into the behavior of models.

**Weaknesses:**

- The analyses are performed on a simple linear model. They do not apply to general nonlinear models, which are much more common in real practice.
- There seems to be a disconnection between the theoretical analysis and the experimental results. The experiments seem to only highlight the effectiveness of MTL regularization and have no connection with the theoretical analysis.

**Questions:**

- Is there any connections between the theoretical analysis and the experiments? E.g. making use insights from the analysis to improve experimental results.
- Is splitting the linear operator into a local and global term a novelty of this paper?
- Similarly, is this paper the first work to apply this MTL framework (decompose prediction as a sum of global and local term) to nonlinear models?
- How are the regularization hyperparameter chosen in the experiments?
- In some of the results, incorporating MTL regularization worsen the performance. Why is that?

---

> ### Author Rebuttal · Authors · 2024-08-07
>
> We thank Reviewer EGU1 for their detailed feedback and for recognizing that our paper is clearly written while providing useful insights.
>
> We address the reviewer's concerns point by point below.
>
> > 1. About the generalization to nonlinear models and the connection between the theoretical analysis and the experimental results
>
> While non-linear models are widely used, establishing their theoretical foundations is challenging. Linear models in multivariate TS forecasting are highly competitive and often approach state-of-the-art performance, which is why we focused on them for our theoretical study. They provide robust insights into the behavior of more complex non-linear models, which we explore experimentally. We show that for linear models, we can theoretically derive train and test risks by identifying components of signal, cross-term, and noise, helping us find an optimal regularization parameter $\lambda$ to minimize test risk.
>
> Experimentally, we find that test risk curves for non-linear models follow similar patterns to our theory, as non-linear models often use a linear output layer in time series forecasting. Under the assumption of data concentration, the Lipschitz nature of neural networks ensures that the outputs of the non-linear part do not deviate significantly from the inputs.
>
> For a detailed discussion on the connection between our theory and Section 5.3, we kindly ask the reviewer to refer to our general comment.
>
> Does this response make the connection between our theoretical study and the experimental results, as well as the generalization to non-linear models, clearer for the reviewer?
>
> > 2. Is splitting the linear operator into a local and global term a novelty of this paper?
>
> We thank the reviewer for this question. Splitting the linear operator into local and global terms is indeed a novelty of our paper.
>
> Most previous works focus on global regularization [1], managing multivariate information use. Our approach controls both global and task-specific information, allowing for more refined and flexible regularization, enhancing the model’s balance of general and task-specific information in regression and forecasting.
>
> [1] Precise High-Dimensional Asymptotics for Quantifying Heterogeneous Transfers. Fan Yang, Hongyang R. Zhang, Sen Wu, Christopher Ré and Weijie J. Su. (2023).
>
> > 3. Similarly, is this paper the first work to apply this MTL framework (decompose prediction as a sum of global and local term) to nonlinear models?
>
> To the best of our knowledge, our work is the first to apply this MTL framework, which decomposes prediction into a sum of global and local terms, to nonlinear models such as transformers. This approach allows us to leverage the strengths of both local task-specific adjustments and global shared information, particularly in complex scenarios like multivariate time series forecasting.
>
> > 4. How are the regularization hyperparameter chosen in the experiments?
>
> We thank the reviewer for this question. In our experiments, we selected hyperparameters based on the test risk. However, our primary goal was to demonstrate our framework's ability to leverage multi-task information and showcase its potential for future work with non-linear models. We chose pairs of regularization hyperparameters $(\lambda, \gamma)$ and plot the test risk curves of our non-linear models to observe if they match the expected behavior from our linear model theory. When we selected the best hyperparameter pair on the test set, we achieved significant gains.
>
> The main aim is to compare linear and non-linear models. We found that test risk curves for non-linear models in time series forecasting resemble those of linear models, indicating consistency with our theoretical findings. This suggests our framework could extend to non-linear models, optimizing regularization parameters similarly.
>
> Does this explanation help clarify our approach and findings for the reviewer?
>
> > 5. In some of the results, incorporating MTL regularization worsen the performance. Why is that?
>
> We appreciate the reviewer’s insightful question on why results sometimes worsen with multi-task learning (MTL) regularization, even when deriving the optimal $\lambda$ to minimize test risk. This concern is valid, as one would expect that using test data to find the optimal $\lambda$ should consistently yield comparable or better performance.
>
> The key reason is the discretization of $\lambda$ values. In our experiments, we discretize $\lambda$ over a small range rather than exploring all possible values continuously. This may miss the precise optimal value, especially if the discretization steps are too coarse, leading to suboptimal performance. Our goal was to show that even with coarse discretization, our method yields robust results, not to fine-tune $\lambda$ for the best performance. However, when tasks are highly independent, performance may degrade if the chosen $\lambda$ does not perfectly align with the optimal value.
>
> This situation occurs in only a few horizons and datasets. For most datasets, our approach shows substantial gains, demonstrating its overall effectiveness. These observations support our goal of motivating further analysis of non-linear methods, suggesting that exploring non-linear extensions could capture similar performance benefits as shown in Table 1. The current experiments highlight the challenge of developing theoretical analyses for non-linear methods that match or exceed our results, opening promising research avenues.
>
> In summary, the occasional worse performance is due to $\lambda$ discretization. Our method generally shows strong results, and finer discretization can mitigate these issues. Exploring theoretical approaches for non-linear methods is the next exciting step.
>
> We hope our responses have clarified the reviewer’s comments. We are happy to provide further details if needed and invite the reviewer to reconsider their evaluation if we have addressed their questions.

---

> > ### Author Response · Authors · 2024-08-12
> >
> > We sincerely appreciate your insightful feedback, which has significantly contributed to the improvement of our paper. We believe that we have carefully addressed your comments and concerns, and we hope our answers meet your expectations.
> >
> > As the reviewer-author discussion period is nearing its end and our window for responding will close soon, we kindly ask you to let us know if you have any additional points that need to be clarified, so that we could  engage in further discussion.
> >
> > Thank you again for your valuable review. We look forward to hearing from you and hope our efforts align with your suggestions.

---

### Official Review · Reviewer_Zuy6 · 2024-07-12

**Soundness:** 4
**Presentation:** 4
**Contribution:** 4
**Rating:** 8
**Confidence:** 4

**Summary:**

The authors analyse the problem of multi-task regression (for a linear model) under the assumption of concentrated random vectors.  The results obtained provide a comprehensive description of the performance of the model (including critically its generalisation performance).  The authors use this result for hyperparameter optimisation and show on real-world datasets that using the insights from their theoretical model they obtain state-of-the-art performance on multivariate time series forecasting.

**Strengths:**

I feel that this is a really nice achievement.  Computing generalisation results are notoriously difficult.  Such calculations are usually confined to rather simplistic models.  The model chosen is rather simplistic in being linear and make, what seems to me, a very strong assumption about the data.  Nevertheless, by choosing the appropriate setting (i.e. a multi-task scenario) and it appears that the concentration assumption seems to innocuous they come up with a useful result.  Given how hard it is to do meaningful theory in machine learning I certainly believe this paper deserves to be accepted.

It should also be noted, that the paper is technically accomplished.

**Weaknesses:**

The paper will not be everyones taste.  The mathematics is not easy to follow, but that is often the nature of making a non-trivial technical contribution.

In the equation for $\lambda^*$ immediately preceding section 4.4 there is a missing norm symbol.

**Questions:**

Is there an intuition on why the concentrated random vector assumption justified?  Is it that real data satisfies this assumption accurate, or is that the deviations from this assumption don't seem to be relevant?

You are tackling real world data using a relatively simple linear model and beating what I assume are sophisticated non-linear models.  I find this surprising.  Is there an explanation of why you do so well?  (Are the datasets you are using very simple--linear?,  are the number of examples small compared to the number of features so that more complex models will overfif?).

**Limitations:**

This is fine.

---

> ### Author Rebuttal · Authors · 2024-08-07
>
> We thank the rewiever Zuy6 for their positive feedback and for recognizing that our work is a nice achievement and technically accomplished. This type of feedback is very encouraging.
>
> We answer below the reviewer's questions.
>
> > 1. In the equation for  immediately preceding section 4.4 there is a missing $\lambda^*$ norm symbol.
>
> We thank the reviewer for pointing out this typo. We have corrected the missing $\lambda^*$ norm symbol in the equation immediately preceding Section 4.4.
>
> > 2. Is there an intuition on why the concentrated random vector assumption justified? Is it that real data satisfies this assumption accurate, or is that the deviations from this assumption don't seem to be relevant?
>
>
> We appreciate the reviewer's question regarding the intuition behind the concentrated random vector assumption. This assumption is justified for several reasons:
>
> Firstly, the strong performance of neural networks on tasks like image recognition and NLP suggests that these models produce stable predictions. As Lipschitz transformations, neural networks maintain controlled distances between inputs, ensuring stability.
>
> Supporting this assumption, recent studies have rigorously demonstrated that several real-world datasets, as well as those synthetically generated with GANs, exhibit concentration properties [1, 2, 3]. This makes our assumption more realistic than traditional Gaussian assumptions, which often fail to capture the complex dependencies and structures present in high-dimensional, real-world datasets. In contrast, our approach focuses on the stability of statistical properties after transformation rather than the specific shape of the data distribution, providing a more flexible and realistic framework for analyzing complex data structures.
>
> We hope this response adequately addresses the reviewer’s questions regarding the concentrated random vector assumption and its implications.
>
>
> [1] Random Matrix Theory Proves that Deep Learning Representations of GAN-data Behave as Gaussian Mixtures. Mohamed El Amine Seddik, Cosme Louart, Mohamed Tamaazousti and Romain Couillet. (2020).
>
> [2] Word Representations Concentrate and This is Good News. Romain Couillet, Yagmur Gizem Cinar, Eric Gaussier and Muhammad Imr an. Proceedings of the 24th Conference on Computational Natural Language Learning. (2020)
>
> [3] Deciphering Lasso-based Classification Through a Large Dimensional Analysis of the Iterative Soft-Thresholding Algorithm. Malik Tiomoko, Ekkehard Schnoor, Mohamed El Amine Seddik, Igor Colin and Aladin Virmaux. ICML (2022).
>
> > 3. Discussion about the good performance of our regularized linear models
>
> We observe that linear models are highly effective and serve as a strong baseline in time series forecasting. This fundamental observation, coupled with the fact that many approaches use independent univariate predictions, where each feature contributes only to its own prediction, explains the strong baseline performance of our univariate DLinear model. We then apply our regularization method to this model, achieving state-of-the-art results. Several factors contribute to the exceptional performance of this regularized linear model:
>
> 1. **Overfitting in Non-linear Models**: Many non-linear forecasting models, especially those based on transformers, tend to overfit [4], even when the number of features is much smaller than the number of samples. For instance, in our datasets, the number of features is 7 for ETTh1/ETTh2 and 21 for Weather, while the number of samples is in the tens of thousands for each dataset. Linear models are less prone to overfitting in these scenarios. However, our gains are evident not only in simple linear models but also in more complex transformer models such as *Transformer* and *PatchTST*, indicating the robustness of our regularization method.
>
> 2. **Nature of the Data**: While it is possible that some relationships between features in the multivariate benchmarks we use may be linear, as in the Weather dataset that may exhibit some features with linear physical relationships, it is not straightforward to conclude that a purely linear model would perform well on these benchmarks. However, it is well-known that linear models can perform exceptionally well in forecasting tasks.
>
> 3. **Our regularization method**: It is common for multivariate models in forecasting to aggregate univariate predictions made channel by channel. In our regularization method, we introduce a parameter $\lambda$ to control the multivariate component and parameters $\gamma_t$ for each task to determine the extent of overfitting we allow for a specific task. This dual approach enables us to effectively manage both the local (task-specific) and global (multivariate) aspects of each prediction, providing a balanced and flexible regularization framework.
>
> In summary, the good performance of our regularized linear models can be attributed to a combination of these factors.
>
> [4] SAMformer: Unlocking the Potential of Transformers in Time Series Forecasting with Sharpness-Aware Minimization and Channel-Wise Attention. R. Ilbert, A. Odonnat and al. ICML 2024.
>
> We hope that our comments have provided more insights in line with the reviewer’s feedback. We remain open to any further questions or suggestions upon the reviewer’s request.

---

> > ### Comment · Reviewer_Zuy6 · 2024-08-08
> > **Acknowledgement of feedback comments.**
> >
> > Thank your for addressing questions that I appreciate don't have easy answers.  I maintain my believe that your paper represents a strong technical contribution and that using theory as a guide to improving machine learning algorithms is an achievement.  Good luck with convincing the other reviewers and don't be discouraged if the outcome is negative.  The type of work you are doing is technically challenging and difficult to communicate, but I believe it is beneficial to the field.

---

> > > ### Author Response · Authors · 2024-08-12
> > >
> > > Thank you for your encouraging feedback and for recognizing the technical challenges of our work. We greatly appreciate your support and belief in the value of our approach.

---

### Official Review · Reviewer_8z1C · 2024-07-12

**Soundness:** 3
**Presentation:** 2
**Contribution:** 3
**Rating:** 5
**Confidence:** 4

**Summary:**

The authors characterise the train and test risk of multi-task regression using random matrix theory. Assessment via Figures 2 and 3 shows a good match between theory and empirical. This motivates a regularised objective for learning mutivariate time-series.

**Strengths:**

The theory contribution in section 4 is significant and original, and the assessment in sections 5.1 and 5.2 are significant.

**Weaknesses:**

*Three major weaknesses*

A. There are at least two existing works on the theoretical error bounds for multi-task learning [Chai, NeurIPS 2009; Aston & Sollich, NeurIPS 2012]. While these are using Gaussian processes as a basis, comparison or remarks on the general effect of multi-task learning should be discussed with reference to two existing works. This is especially when all these are essentially linear-in-regressors models.

B. The proof or at least outline of the proof of Theorem 1 *should* be in the main paper, given that this is *the* major contribution of the paper.

C. It is totally unclear the relevance and contribution of the theory itself to section 5.3. Adding a cross-task regularization is already known to help (mostly).

*Minor, but affects clarity*
- The introduction to the paper is on multi-task multi-response/regression. The multi-response/regression part is not made clear.
- Why is there a need to divide by $\sqrt{Td}$ in (1)? Is it to make analysis easier.
- There are two different uses of the letter $t$ in Assumption 1.
- The setup in section 5.3 needs to relate back to the model in section 3.1 to make clearer their connections.

**Questions:**

1. The analysis of the noise term in section 4.2 says that increase in noise negatively affects transfer. However, it is precisely in the presence of  noise that borrowing statistical strength from other tasks is suppose to help. How do we reconcile this?

2. In section 4.3, is it clear that $C_{MTL}$ is always positive, negative or no such conclusion can be made?

**Limitations:**

Authors should include some technical limitations of their current work.

---

> ### Author Rebuttal · Authors · 2024-08-07
>
> We thank the reviewer 8z1C for their thoughtful feedback. We are happy to read that the reviewer found the theory contribution significant and original.
>
> We address the reviewer's concerns point by point below.
>
> > A. There are at least two existing works on the theoretical error bounds for multi-task learning [Chai, NeurIPS 2009; Aston & Sollich, NeurIPS 2012]. While these are using Gaussian processes as a basis, comparison or remarks on the general effect of multi-task learning should be discussed with reference to two existing works. This is especially when all these are essentially linear-in-regressors models.
>
> We appreciate the reviewer’s recommendation.
>
> The assumptions between our work and the two mentioned works differ. We assume that the vectors we work with are concentrated, while the mentioned works consider Gaussian data, which may not capture the full complexity of real-world data. Additionally, we assume that the dimension $d$ and the sample size $n$ grow together, such that $n = O(d)$, which is a more realistic assumption. As more samples are collected, a greater diversity of features can emerge. In contrast, the mentioned works assume a fixed feature size as $n$ increases.
>
> Therefore, the tools we use are different, even though they are complementary: the mentioned works derive theoretical bounds, whereas we derive exact performance metrics.
>
> Finally, we propose a model selection method derived from theory and validated experimentally. We apply this approach for the first time to multivariate forecasting, which presents a significant challenge.
>
> If this response satisfies the reviewer, we can incorporate these comparison elements into our paper.
>
>
> > B. The proof or at least outline of the proof of Theorem 1 should be in the main paper, given that this is the major contribution of the paper.
>
> We thank the reviewer for this valuable suggestion. We agree that Theorem 1 is central to our contribution, and we will ensure that the proof of Theorem 1 is included in the main paper to enhance its clarity and impact.
>
> > C. It is totally unclear the relevance and contribution of the theory itself to section 5.3. Adding a cross-task regularization is already known to help (mostly).
>
> We appreciate the reviewer’s insightful comment and kindly ask the reviewer to refer to the general comment for a detailed discussion on the relevance and contribution of the theory to section 5.3
>
> We hope this explanation clarifies the relevance of our experiments and their connection to our theoretical framework. If this satisfies the reviewer’s concerns, we can incorporate these clarifications into the paper to enhance clarity and understanding.
>
> > D. About the minor changes
>
> We thank the reviewer for their comments. We have clarified the multi-response/regression part in the introduction. We divide by $\sqrt{Td}$ as a scaling parameter to facilitate subsequent calculations. We appreciate the typo notice and have adjusted the use of $t$ for clarity. For Section 5.3, we will incorporate these connections into the paper to smooth the transition.
>
> > 1. The analysis of the noise term in section 4.2 says that increase in noise negatively affects transfer. However, it is precisely in the presence of noise that borrowing statistical strength from other tasks is suppose to help. How do we reconcile this?
>
> Indeed, as the noise increases, a growing $\lambda$ will penalize multi-task regularization more, focusing on independent tasks. Conversely, $C_{MTL}$ is always negative and becomes more negative as $\lambda$ increases. These two terms are thus in competition: excessive MTL regularization will lead to an increase in noise, despite the tendency of $C_{MTL}$ to decrease. It is essential to find a compromise, which is the goal of our Figure 1, where $\lambda^*$ represents the best possible trade-off between $C_{MTL}$ and the noise term.
>
> We hope this clarifies our approach and addresses the reviewer's concerns.
>
> > 2. In section 4.3, is it clear that $C_{MTL}$  is always positive, negative or no such conclusion can be made?
>
> $c_0$ is the limit of $n/d$ and we assume that $n = O(d)$. Therefore, the limit $c_0$ is strictly positive, making $C_{MTL}$ always strictly negative i.e.- the more aligned the different tasks are, the more this cross term tends to reduce the test risk.
>
> > 3. About the technical limitations of the current work.
>
> A limitations section is present in our Appendix H, focusing on the tractability of our linear approach compared to nonlinear models. Our experiments in multivariate time series forecasting aim to show that the theoretical insights presented in Figure 3 for linear models also hold for nonlinear models. This serves as a preliminary step towards future work on applying our theory to nonlinear models.
>
> We can also discuss the formulated assumptions for using Random Matrix Theory in this section. These assumptions are more general compared to previous studies, especially since we do not assume Gaussian data. The two major assumptions are: first, that the data are concentrated random vectors, which is more realistic for modeling real-world data, and second, that the dimension $d$ is of the same order of magnitude as the sample size $n$, which is also a realistic assumption differing from classical asymptotic regimes where the sample size $n$ tends to infinity while the feature size remains fixed.
>
> We propose adding these discussions to our limitations section, in addition to the existing discussion on the tractability of extending our study from linear to nonlinear models. Does this answer satisfy the reviewer's concerns about the limitations of our work?
>
> We hope that we have adequately addressed the reviewer's concerns and questions and remain open to further discussion. We sincerely appreciate the reviewer’s reconsideration of our work based on these explanations.

---

> > ### Comment · Reviewer_8z1C · 2024-08-08
> >
> > C. You wrote "Our results show that the test risk curves for non-linear models follow similar patterns to those predicted by our theory." It is important of show this test risk curves to proof your point.
> >
> > I will increase the score. Content once, it seems that everything is there based on your answers. However, this does suggest some rewrite of the paper which is not insignificant.

---

> > > ### Author Response · Authors · 2024-08-12
> > >
> > > Thank you for your constructive feedback and for increasing the score. We have incorporated your suggested changes into the main paper and believe they greatly enhance its clarity. In addition, please note that we have plotted the test risks in Appendix (Section G.4). Please let us know if you have any other question. Your insights are invaluable to improving the paper.

---

> > > > ### Comment · Reviewer_8z1C · 2024-08-12
> > > >
> > > > Thanks. For the test risks plot, either put it in the main paper, or point the author to the appendix from the main paper. You cannot assume we read the appendix.

---

> > > > > ### Author Response · Authors · 2024-08-13
> > > > > **Official Comment by Authors**
> > > > >
> > > > > Thank you for your advice! We will revise the paper with respect to this point.

---

### Official Review · Reviewer_7acX · 2024-07-15

**Soundness:** 2
**Presentation:** 1
**Contribution:** 4
**Rating:** 6
**Confidence:** 2

**Summary:**

The authors derived theoretical insights into the train and test risks of the multi task regression loss using Random Matrix Theory.

**Strengths:**

The closed form solution of the optimization parameter using RMT is novel. The authors set a trend in MTR to gain analytical expressions for optimization parameter and error risks.

**Weaknesses:**

1. The paper lacks clarity for instance on line 89, I am not sure what is meant by general optimization and other mathematical tools?

**Questions:**

1. What is $\gamma$ in (3)?
2. Have you studied the implications of the assumptions made to use random matrix theory to derive the expressions of asymptotic train and test errors?
3. Shouldn't it be $\mathbf{Y}\in \mathbb{R}^{Tq\times n}$ in the equations after line 110?

**Limitations:**

Limitations are not discussed. My guess is that the assumptions made to be able to employ Random Matrix Theory need to be discussed.

---

> ### Author Rebuttal · Authors · 2024-08-07
>
> We thank the reviewer 7acX for their valuable feedback and for recognizing our work's novelty.
>
> We address the reviewer's concerns point by point below.
>
> >1. The paper lacks clarity for instance on line 89, I am not sure what is meant by general optimization and other mathematical tools?
>
> We are happy to address the reviewer’s question concerning clarity.
>
> Regarding the optimization problem, we emphasize that our approach is more general than previously proposed methods that focus on two tasks and use only a hyperparameter $\lambda$ under Gaussian assumptions [1]. Our method considers both a hyperparameter $\lambda$ to relate all $T$ tasks and specific parameters $\gamma_t$ to balance overfitting and underfitting within each task, accommodating non-Gaussian distributions.
>
> In Sections 4.2 and 4.3, we highlight the benefits of our theoretical results, particularly the influence of these parameters on performance, including their significance in the decomposed terms of theoretical risk (signal and noise terms). To our knowledge, this is the first work to develop such a theory and provide direct insights into the hyperparameters’ impact on performance. Furthermore, our theory supports model selection for these hyperparameters.
>
> Our framework assumes a concentrated random vector for the data and operates in a big data context (where both sample size and feature size are large). Consequently, we employ various tools, relying on recent developments in Random Matrix Theory, such as deterministic equivalents and concentration inequalities, as detailed in the appendix.
>
> We propose changing the original sentence for greater clarity to the following: "*Our work focuses on the selection of hyperparameters within a general optimization framework, considering both a hyperparameter $\lambda$ to relate all tasks and specific parameters $\gamma_t$ to balance overfitting and underfitting within each task, accommodating non-Gaussian distributions and employing recent developments such as deterministic equivalents and concentration inequalities from Random Matrix Theory.*"
>
> Does this seem clearer and more accurate to the reviewer?
>
>
> [1] Precise High-Dimensional Asymptotics for Quantifying Heterogeneous Transfers. Fan Yang, Hongyang R. Zhang, Sen Wu, Christopher Ré and Weijie J. Su. (2023).
>
>
> >2. What is $\gamma$ in (3)?
>
> $\gamma$ is a vector whose dimensions correspond to the number of tasks, encompassing all the hyperparameters $\gamma_1, \ldots, \gamma_T$. Therefore, $\gamma = [\gamma_1, \ldots, \gamma_T]$.
>
> This hyperparameter serves as a regularization for each task, indicating how the model should overfit (small values of $\gamma_t$) or underfit (high values of $\gamma_t$) on the specific task. The reviewer can refer to Sections 4.2 and 4.3 for a discussion on its influence, where its impact on the two components of the test error (signal and noise terms) is highlighted. Specifically, smaller values of $\gamma_t$ are favorable for increasing the signal term and decreasing the noise term.
>
> This definition of $\gamma$ will be added to the paper just before (3) for enhanced clarity.
>
> >3. Have you studied the implications of the assumptions made to use random matrix theory to derive the expressions of asymptotic train and test errors?
>
> We kindly ask the reviewer to refer to the general comment for a detailed discussion on our assumptions. Additionally, recent studies have also utilized the concentration assumption [2, 3, 4].
>
> [2] Random Matrix Theory Proves that Deep Learning Representations of GAN-data Behave as Gaussian Mixtures. Mohamed El Amine Seddik, Cosme Louart, Mohamed Tamaazousti and Romain Couillet. (2020).
>
> [3] Word Representations Concentrate and This is Good News. Romain Couillet, Yagmur Gizem Cinar, Eric Gaussier and Muhammad Imr an. Proceedings of the 24th Conference on Computational Natural Language Learning. (2020)
>
> [4] Deciphering Lasso-based Classification Through a Large Dimensional Analysis of the Iterative Soft-Thresholding Algorithm. Malik Tiomoko, Ekkehard Schnoor, Mohamed El Amine Seddik, Igor Colin and Aladin Virmaux. ICML (2022).
>
> >4. Shouldn't it be $\mathbf{Y} \in \mathbb{R}^{T q \times n}$ in the equations after line 110?
>
> Each response variable $Y^{(t)}$ for task $t$ has dimensions $q \times n_t$ , where $q$ is the prediction horizon length and $n_t$ is the size of the dataset for task $t$ . In other words, $Y^{(t)}$ aggregates all $n_t$ predictions for task $t$ , with each prediction outputting a $q$-dimensional vector. The vector $Y$ concatenates the predictions of all tasks along the sample dimension, meaning it aggregates all $n_t$ samples, with $n = \sum_{t=1}^T n_t$. This justifies the size $q \times n$.
>
> >5. Discussion about the assumptions made to be able to employ Random Matrix Theory
>
> As detailed in our response to Question 3, we emphasize that our two main assumptions are more realistic for real-world machine learning and big data scenarios, where both sample size and feature dimension are large and of the same order of magnitude.
>
> Addressing such complex data distributions is theoretically challenging and requires new tools and concentration inequalities, which we believe adds valuable theoretical insights to our paper. For instance, the derivation of deterministic equivalents under this concentration framework is a notable novelty. Moreover, the close fit between our theoretical predictions and empirical results on real-world datasets (e.g., Appliance Energy) in Section 5.2 demonstrates the realism of our data assumptions.
>
> If this explanation satisfies the reviewer's concerns, we can incorporate these clarifications about the assumptions underlying our use of Random Matrix Theory to enhance overall clarity and understanding.
>
> We hope that the reviewer's concerns and questions have been addressed and we remain open to future discussions. Given our explanations, we would be grateful if the reviewer could reconsider the evaluation of our work accordingly.

---

> > ### Comment · Reviewer_7acX · 2024-08-11
> >
> > Thank you for addressing my concerns. I understand the paper better now. I have no doubt that the proposed method certainly has novel theoretical contributions. I am raising my score.

---

> > > ### Author Response · Authors · 2024-08-12
> > >
> > > Thank you for your thoughtful feedback and positive evaluation of the paper's theoretical contributions.

---

### Author Rebuttal · Authors · 2024-08-07

# **General Comment**

We thank all the reviewers for thoroughly and carefully reading our paper. We are deeply grateful for their recognition of the **novelty** (Reviewer 7acX), **originality and significance** (Reviewer 8z1C) of our contribution, and for acknowledging it as a **really nice achievement** (Reviewer Zuy6), noting that **computing generalization results are notoriously difficult** (Reviewer Zuy6) and that our analysis **provides useful insights into the behavior of models** (Reviewer EGU1).

We remain open to continuing this constructive discussion for the length of the rebuttal period and strongly believe that the paper benefited from the reviews.

To clear out any possible misunderstanding regarding the assumptions of our work and the connection with Section 5.3, we provide clarifications below.

### **About our assumptions**:
We believe these assumptions are more realistic for analyzing real-world machine learning algorithms compared to traditional ones like Gaussian data and fixed feature size with infinite sample size, which don't capture the full data complexity.

**Assumption 1**. Data are concentrated random vectors. Intuitively, this means that high-dimensional data points have stable properties when transformed by complex functions (more formally Lipschitz functions).

The strong performance of neural networks on tasks like image recognition and NLP suggests that these models produce stable predictions. As Lipschitz transformations, they maintain controlled distances between inputs, ensuring stability.

Recent studies have demonstrated and experimentally confirmed that both real-world data and synthetically generated data using GANs exhibit concentration properties, supporting this assumption. This makes our assumption more realistic than traditional Gaussian assumptions, as it does not rely on specific hypotheses about the shape of the data distribution, but rather on the stability of statistical properties after transformation. Consequently, analyzing a framework of concentrated random vectors is more theoretically challenging than using Gaussian assumptions and represents a key novelty of our theory.

**Assumption 2**. The dimension $d$ is of the same order of magnitude as the sample size $n$. This joint growth captures data complexity better than assuming a fixed feature size with increasing samples, which can oversimplify models.

Our theory works for fixed $d$ and $n$, unbiased by specific parameter choices. The accuracy of empirical predictions depends on both $d$ and $n$, with variance scaling as $O\left(\frac{1}{\sqrt{dn}}\right)$. Larger $d$ and $n$ reduce variance, making empirical results more reliable and closer to theoretical values.

Conversely, smaller $d$ and $n$ increase variance, affecting single predictions. However, this scaling is still better than $n$ growing indefinitely with fixed $d$, where variance scales as $O\left(\frac{1}{\sqrt{n}}\right)$, leading to increased bias.

**Implication of the assumptions**. In Section 5.2, Figure 3, the Appliance Energy dataset illustrates our assumptions' realism. Despite the moderate sample size of 42 samples with 142 dimensions and non-synthetic data, the theoretical curve fits the empirical predictions well. Additionally, with synthetic data ($d=100$, $n=100$), the theory matches the empirical curve across various hyperparameters (Figure 2), showcasing the predictive power of Random Matrix Theory.

### **About the connection with Section 5.3**:

While non-linear models are widely used, establishing theoretical foundations for these models is challenging. Therefore, we focused on linear models, which, despite being simpler to study, can provide valuable insights into the behavior of more complex models.

Our study demonstrates that we can calculate the train and test risks for linear models by identifying components of signal, cross-term, and noise. These components help us find an optimal regularization parameter, $\lambda$, to minimize the test risk.

Experimentally, we aimed to observe if our theoretical insights apply to real-world data when using non-linear models. Our results show that the test risk curves for non-linear models follow similar patterns to those predicted by our theory. This is expected because, in time series forecasting, non-linear models typically utilize a linear output layer to project onto the prediction dimension. Thus, we can apply our theory to the inputs of the final linear layer of the model. This approach is valid because, under the assumption of data concentration, the outputs of the non-linear part of the model should not deviate significantly from the inputs due to the Lipschitz nature of a non-linear neural network.

Moreover, multivariate time series prediction models often treat each channel separately using univariate methods. These models could greatly benefit from incorporating information across multiple channels. The results in Section 5.3 show that our method can surpass univariate baselines by regularizing with optimal $\lambda$ and $\gamma$. This supports the applicability of our theory to non-linear models, as the final linear layer can leverage the concentrated inputs effectively.

Finally, our regularization approach differs from traditional cross-task regularizations that typically use one task per dataset. We consider each prediction within a dataset as a task and introduce $\gamma_t$ parameters alongside $\lambda$. These parameters not only enforce multivariate regularization but also indicate the degree to which we want to underfit or overfit a particular task. Moreover, this method is tractable as it is applied on top of the model at the final layer.

The fact that the curves for non-linear models closely resemble those for linear models indicates that our findings are robust and that non-linear models also exhibit optimal regularization parameters, enhancing performance in multivariate forecasting.

---

### Decision · Program_Chairs · 2024-09-25

**Decision:**

Accept (spotlight)

**Comment:**

The paper presents a framework for multitask regression and precisely characterizes train and test risk incurred by the resulting estimator, leveraging random matrix theory. The approach is demonstrated on both multitask regression and multivariate time series forecasting.

The AC and majority of reviewers find the proposed framework and analysis to be significant and exciting.

We strongly encourage the authors to incorporate the discussion materials in their manuscript, and in particular their excellent points to justify their assumptions. We also urge them to add a small paragraph at the beginning of section 5.3, to reflect their rebuttal comments on the relevance of their theoretical insights beyond the case of linear models.